



Summer-time episodic chlorophyll-a blooms near east coast of Korea
Young-Tae Son, Jae-Hyoung Park, and SungHyun Nam[*]
Seoul National University, Seoul, Republic of Korea
[*]Correspondence to namsh@snu.ac.kr



## Abstract

We present intensive observational data of surface chlorophyll-a bloom episodes occurring over several days in the summers of 2011, 2012, and 2013, accompanying the equatorward advection of low sea-surface salinity (SSS) water near the east coast of Korea. Time-series analysis of meteorological and oceanographic (physical and biochemical) parameter data, such as chlorophyll fluorescence (CF) from surface mooring, ocean color (chlorophyll a and total suspended sediment), sea surface height (satellite-derived), and serial hydrographic data (from in-situ measurements) were used to investigate the relationship between surface bloom events and changes in seawater characteristics and currents. In the summers of the three years, a total of 10 bloom events (E01–E10) were identified where the surface CF was significantly ($> 2$ μg/$l$) enhanced over a relatively long ($> 1$ day) period. The bloom events in the summers of 2011 and 2012 accompanied low or decreasing SSS for several days to a week after heavy rainfalls at upstream stations and equatorward currents. Unlike the typical 8 of the 10 events (80 %), E07 was potentially derived from the onshore advection of high CF offshore water of southern origin into the coastal zone near the mooring, whereas E10 is likely prevailed by offshore advection of high CF plume water trapped by the coastal area. Contrasting with many coastal systems, these findings indicate that event-scale productivity near the east coast of Korea in summer is not controlled by local blooms triggered by either nutrients or light availability, but by the equatorward and cross-shore advections of high CF plume water.



## 1. Introduction

Biological blooms associated with, among others, the horizontal advection of chlorophyll-rich water (often having low-salinity and high nutrients linked to heavy rain, e.g., nutrient loading), have been frequently observed in many coastal systems (e.g. Yin et al., 2004; Dai et al., 2008; Halverson and Pawlowicz, 2013; Reifel et al., 2013). Blooms stimulated by plume-delivering nutrients and enhanced stratification were observed near and offshore of Hong Kong (Dai et al., 2008; Yin et al., 2004). During bloom events, a several-fold increase in chlorophyll a (Chl a) and significant shift in phytoplankton community structure were observed (Dai et al., 2008). The effects of effluent discharge plume on coastal phytoplankton communities were examined from the City of Los Angeles Hyperion Wastewater Treatment Plant, demonstrating localized blooms occurred a few days after the diversion within the effluent plume (Reifel et al., 2013). The Fraser River plume affects Chl a distribution in the Strait of Georgia, British Columbia, Canada, revealing large differences with respect to the plume, despite insensitivity in the long-term average (Halverson and Pawlowicz, 2013).

There are several small river plumes potentially affecting Chl a distribution near and offshore of the east coast of Korea; yet, the effects remain poorly understood. High summer (from June to September, JJAS) precipitation often accompanying heavy rainfall around the Korean peninsula is well known and accounts for more than 50% of the annual precipitation in the region. During summer, most rivers in the region become flooded and discharge large volumes of freshwater into the adjacent marginal seas, including the East Sea (Japan Sea), Yellow Sea, and East China Sea (Bae et al., 2008; Kong et al., 2013). Chl a distribution in the southwestern East Sea off the east coast of Korea has been examined, and found to be associated with physical processes at mesoscale or larger scales, including spring and fall blooms that have been detected using satellite ocean color data, data from limited short-duration ship surveys (Hyun et al., 2008;Kang et al., 2004), and time-series data collected continuously from moored buoys (Hong et al., 2013;Son et al., 2014). Despite wide range images available from geostationary and polar-orbit satellite ocean color remote sensing (Yoo and Kim, 2004;Son et al., 2014;Hyun et al., 2008;Kim et al., 2011), phytoplankton blooms observed over several days to weeks near the coast, particularly during the well-stratified summer season, have rarely been examined. Thus, we aimed to address the episodic bloom events in summer and investigate the effects of river plumes on Chl a distribution near and away from the east coast of Korea.

## 2. Data and methods

Time-series data of meteorological, physical, and biochemical parameters have been measured using a surface mooring named ESROB (East Sea Real-time monitoring Ocean Buoy), deployed in water at 130 m depth, about 8 km off the mid-east coast of Korea (Fig. 1). The data collected includes wind speed and direction at 2 m above the sea surface, photosynthetically active radiation (PAR) at about 2 m above the sea surface and at 10 m depth, temperature and salinity at five depths (5 m, 20m, 40 m, 60 m, and 110 m), vertical profile of current with an interval (bin size) of 4 m (upper most bin corresponds to 5 m depth), and sea surface



temperature (SST), salinity (SSS), dissolved oxygen (DO), and chlorophyll fluorescence (CF) measured by a Water Quality Monitor (WQM) at about 1 m depth. Details on the technical design, improvements, and early-phase operations of ESROB have been previously described (Nam et al., 2005). In the present study, we used data collected for ~ 3 years, from April, 2011 to December, 2013, with an emphasis on the three summer periods (JJAS) when the alongshore current averaged over 6 years reversed to an equatorward direction (Park et al., 2016).

The CF as a factory-calibrated Chl a concentration in units of μg/l following the manufacturer's (WET Lab) instructions is needed to calibrate with in-situ measurements owing to long-term sensor drift. Four cruises were conducted in July and October 2011, April 2012, and July 2013 to collect in-situ water samples for Chl a and in-situ sensor measurements for water temperature and salinity near the coast. A statistically significant correlation ($r^2 = 0.76$, $p < 0.001$) was found between the CF sensor values and in-situ chlorophyll concentration derived from the spectrophotometer using acetone-extracted Chl a (Fig. 2a). In addition to the chlorophyll calibration, the concentrations of nitrate were analyzed simultaneously with 64 samples to determine the nitrate proxy based on the relationship between temperature and nitrate. Separately, to observe the fine-scale coastal SST and SSS distributions around the ESROB, in-situ measurements using a small research vessel equipped with a thermosalinograph (SEB21, 10 s sampling interval) were conducted on July 30, 2013, a couple of days after heavy rainfall. Since non-photochemical quenching (NPQ) has a significant influence on the CF in response to changes in ambient light (Müller et al., 2001), particularly for a single channel excitation Chl a fluorometer, the effects were corrected from the ESROB CF data following the methods described in Halverson and Pawlowicz (2013) before calibrating with in-situ water samples.

We used high-resolution daily data generated by the geostationary ocean color satellite (composited using eight images) to estimate surface Chl a distributions. The spatial resolution of the geostationary satellite is 500 m at a grid 50 times further than previous polar orbiting ocean color satellites (Ryu et al., 2012). Chl a concentration observed from the ocean color satellite can be easily contaminated by the total suspended sediment (TSS) and colored dissolved organic matter (CDOM) in the coastal regions (Ryu et al., 2012). Thus, the satellite-measured Chl a was calculated through software modules applying a correction algorithm for the TSS and CDOM, as well as by minimizing the contaminating effects of cloud, sea fog, and aerosols (level 1B). Nevertheless, relationships between the satellite-measured Chl a and TSS in coastal and offshore areas in Fig. 1 were compared with a linear regression to determine the Chl a in the coastal region (Fig. 2b, c). Results exhibited that the higher the value, the wider the scatter. This indicated that Chl a can be measured regardless of the TSS both in the coastal and outer sea, which supports the possibility of using the satellite-derived Chl a in this area. Satellite altimeter-derived sea surface height (SSH) products corrected using coastal tide-gauge sea level data along the east coast of Korea (Choi et al., 2012) were used to examine surface geostrophic currents around and offshore of the ESROB in the summer of 2013. Precipitation data were also used to compare the bloom timings with those of heavy rainfalls in summer. Precipitation in unit of mm/day recorded every 3 hours at stations during the summers of 2011, 2012, and 2013 along the coast (SP: SinPho, HH: HamHeung, WS: WonSan, JJ: JangJun, SO: Sockcho, BGN: BukGangNeung, DH: DongHae) were proxied as freshwater discharges from several small rivers into the East Sea (Fig. 1).



Current and wind vectors were corrected for local magnetic deviation, decomposed into alongshore and
cross-shore components rotating counter-clockwise from the north by 30 degrees. Wind stresses have been
calculated following $\vec{\tau} = \rho_a C_D |W| \vec{W}$ ($\rho_a$: air density, $C_D$: drag coefficient, $\vec{W}$: wind), and alongshore and
cross-shore components of current ($V_a$ and $U_c$) and wind stress ($WS_a$ and $WS_c$) are expressed by the
coordinate transformation, respectively (Large and Pond, 1981). All variables were low-pass filtered with
the half power centered at 40 h.

**3. Results**
**3.1. Climatological CF variations**
Annual cycles of wind stress ($WS_a$, $WS_c$), surface CF, SST, SSS, surface DO, and surface current (($V_a$, $U_c$)
at the upper most bin) observed at the ESROB were obtained by climatologically averaging monthly mean
values over the three years from 2011 to 2013, which showed significant summer-time CF enhancements (in
addition to two well-documented blooms in spring and fall), weakened wind forcing, increased SST,
decreased SSS, over-saturated surface DO (though absolute DO decreased), and strengthened equatorward
($V_a < 0$) surface currents (Fig. 3). The enhancements of CF during the summer with significantly high
concentrations over 1 $\mu$g/l in July accompanied with decreased SSS (abruptly decreased from June to July)
and strengthened equatorward currents (maximum speed of 15 cm/s in July), implied high Chl a and low
salinity water of northern origin. Although absolute DO decreased with increasing SST, the surface water
was over-saturated for most of the summer, implying a significant role of surface bioactivity. Weak poleward
($V_a > 0$ and $U_a \sim 0$) surface currents were observed throughout the year, except in summer, when strong
equatorward ($V_a < 0$ and $U_a \sim 0$) currents prevailed.

**3.2. CF events observed in summers of 2011, 2012, and 2013**
In the summers of 2011, 2012, and 2013, 10 bloom events (E01–E10) were identified where the surface
CF was significantly enhanced over considerable period (Fig. 4, Table 1). The CF bloom events were defined
as follows: the peak CF reached higher than 2.0 µg/l and the duration when the CF > 2.0 µg/l was longer than
1 day, when CF > 1.0 µg/l. The summer bloom event lasted for several days to weeks, which is shorter than
the typical duration of spring and fall blooms. Six events, three each year (E01–E03 and E04–E06), were
identified in the summers of 2011 and 2012, whereas four (E07–E10) occurred in 2013 (Fig. 4). The average
SST, SSS, and CF for the duration of each event are listed in Table 1.
During the CF events in the summer of 2011 (E01–E03), low SSS was observed at the ESROB several
days to a week after remarkable wind forcing and heavy rainfalls (maximum of 160 mm/day during E02) at
upstream stations, accompanying enhanced equatorward currents (Fig. 5a, c, e and f). Two typhoons (MAON
and MUIFA) yielding a maximum wind stress of 0.25 N/m² passed through the region during the CF bloom



events, inducing strong equatorward (before E01) and poleward (after E03) wind stresses (arrows labeled by
M1 and M2 in Fig. 5b). Interestingly, the equatorward (poleward) wind stress may strengthen equatorward
(poleward) and onshore (offshore) surface currents. Indeed, strong equatorward currents were observed up
to 2 days after the peak wind forcing immediately before E01, whereas the equatorward currents were
markedly weakened by the poleward wind stress immediately after E03 (Fig. 5b, f).
Similarly, the CF events in the summer of 2012 were also accompanied by low or decreasing SSS several
days to a week after heavy rainfalls at upstream stations and equatorward currents (Fig. 6a, c, e and f). Three
(KHANUN, BOLAVEN, and TENBIIN) among the four typhoons in the summer affected the surface CF,
SSS, and surface currents during the events. Since typhoon KHANUN drove poleward wind stress, strong
equatorward currents developed before E04 were weakened, and SSS increased to reduce the salinity
stratification and decrease surface CF during E04 (arrow labeled by K in Fig. 6b, c, e and f). After the typhoon
passed, the surface CF increased again along with re-enhancing equatorward currents, re-stratifying salinity,
and decreasing SSS during E05 (Fig. 6c, e and f). Two typhoons (BOLAVEN and TENBIIN) successively
passed the area and both poleward (equatorward) wind stress re-stratified (well-mixed) upper ocean
conditions during E06. The poleward wind stress imposed by the BOLAVEN induced well-mixed conditions
with high SSS, low SST, and strong poleward surface currents (arrow labeled by B in Fig. 6b, c, d and f).
However, the reversed wind stress imposed by the successive TENBIN resulted in decreasing SSS, increasing
SST, weakening the poleward surface current (strengthening equatorward surface current), and rapidly
increasing surface CF (peak exceeding 4.5 $\mu$g/$l$) (arrow labeled by T in Fig. 6b, c, d, e and f).
Contrasting to the CF bloom events in the summers of 2011 and 2012, two among the four events (E07
and E10) in the summer of 2013 did not accompany preceding heavy enough rainfall at the upstream stations
nor equatorward currents (Fig. 7a, f). Typical heavy rainfalls and enhanced equatorward surface currents
preceded low SSS and high surface CF during the other two events (E08 and E09) only (Fig. 7a, f). Unlikely
with typical events, the SSS remained high and SST temporally decreased (negative anomaly) during E07
(Fig. 7c and 7d), whereas relatively high SST and low SSS were observed during E10 (Fig. 7c, d). Contrasting
with those in the other two years, winds were mild and no typhoon passage was reported in the summer of
2013 (Fig. 7b).

**3.3. Surface CF distributions**
The equatorward advection of low salinity, chlorophyll-rich plume water into the ESROB area along the
coast was confirmed from a series of daily composite satellite-measured Chl a only when clear images
containing few clouds were available. One example presented here is from four images continuously
available from July 24 to 27, 2013, before E09 (Fig. 9a, b, c and d). A high surface CF zone in the northern
area (e.g. off the SP, HH, and WS, Fig. 1) was separated from that in the southern area (e.g. between the
coast and UI, Fig. 1) following the poleward current—the East Korea Warm Current (EKWC)—and
extended equatorward with time near the coast during the period (Fig. 9a, b, c and d) after the heavy rainfalls





in July 19–24 (Fig. 7a). The high CF plume water was elongated and reached to JJ by July
25–26, and farther south near the coast by July 27, yielding the E09 event from July 28 to August 1 (Table
1, Fig. 7). The SST and SSS observed using the thermosalinograph on July 30, 2013 in the vicinity of ESROB
consistently demonstrated wedge-shaped patterns with low SSS and high SST water confined near the coast
and reaching farther south passing BGN (Fig. 9e and 9f), confirming the equatorward advection of low-
salinity and high CF surface water along the coast to ESROB. Interestingly, the satellite-based surface
geostrophic currents around and offshore of the ESROB (not shown) and the alongshore currents observed
at the upper depths of the ESROB (e.g. Fig. 7f) were all equatorward during this period.
A pattern of surface CF distribution and geostrophic flow field on July 3, 2013 for E07 are shown in Fig.
9a and 9b, where high CF was found inshore of the poleward flowing EKWC (main axis is closer to UI than
the high CF area) and within cyclonic circulation around the ESROB (area of relatively low SSH). Onshore
currents prevailed between BGN and DH, associated with the cyclonic circulation (Fig. 9b), potentially
yielding onshore advection of high CF offshore water of southern origin into the coastal zone near the
ESROB during E07 (Fig. 9d). Similarly, although clear images were not available at that time, the
geostrophic flow field on August 21, 2013 for E10 is shown in Fig. 9c, wherein offshore currents were found
to prevail near the coastal zone. The offshore advection of coastal plume water of northern origin presumably
having low salinity, high temperature, and high CF (as cases of many other events, see Fig. 1 or Fig. 8) may
have enhanced the surface CF at the ESROB during E10 (Fig. 9e).

## 199    4. Discussion

### 200    4.1. Horizontal advection

The low-salinity chlorophyll-rich water originating from the northern coastal region often accompanying
heavy rainfalls is advected equatorward along the coast into the coastal zone in the vicinity of the ESROB in
summer, and is primarily responsible for most (80 %) of the CF events. The rate of Chl $a$ change observed at
the ESROB is comparable with the rate estimated from the spatial Chl $a$ gradient and speed of equatorward
advection. The equatorward advection distance of high Chl $a$ water is measured to 100 km (= dy) over 3 days
(= dt) with Chl $a$ change of about 2.5 μg/l (= dChl) from the series of four daily composites of satellite-
measured Chl $a$ collected in July 24 to 27, 2013 before E09 (Fig. 9a, b, c and d). With an advective speed of
0.4 m/s (= 100 km / 3 days), this yields a rate of Chl $a$ change of 0.86 μg/l d$^{-1}$ (= 0.4 m/s × 2.5 μg/l / 100 km)
owing to the alongshore advection ($v\partial$Chl $a/\partial y$), which is consistent with the observed rate ($\partial$Chl $a/\partial t$
where dChl was estimated from the ESROB measurements and dt =1 h) for E09 (up to 1.26 μg/l d$^{-1}$ averaged
over the period when $\partial$Chl $a/\partial t > 0$) and others (mean: 0.87 μg/l d$^{-1}$), supporting that the alongshore
advection plays a primary role in CF variability near the coast. These findings are similar to those of bloom
events with a rate of CF change (2–4 μg/l d$^{-1}$ estimated from their Fig. 11) controlled by the advection of low
SSS and high CF plume water in other coastal systems (Halverson and Pawlowicz, 2013).
In contrast to E09, the high surface CF observed during E07 is not explained by equatorward advection of





low-salinity chlorophyll-rich water originating from the northern coastal region, but potentially by the
onshore advection of high CF water of southern origin advected via the EKWC. Hyun et al. (2009)
demonstrated that the highest primary productivity in the southwestern East Sea is induced by the
transportation of high CF water originated from upwelling of nutrient rich water along the southern east coast
of Korea. The high CF water may affect the productivity near the mid-east coast of Korea as advected by the
EKWC and its meanders, particularly on the western or coastal side of the front formed by the EKWC. Indeed,
a rate of cross-shore Chl a change around ESROB from the surface CF distribution observed during E07 (Fig.
9a) is roughly 0.1 μg/l km$^{-1}$ (dChl = 1.0 μg/l and dx = 10 km) and a rate of Chl a change by cross-shore
advection (u $\partial$Chl $a/\partial x$) is estimated to 0.86 μg/l d$^{-1}$ (= 0.1 m/s * 1.0 μg/l / 10 km) with cross-shore velocity
of 0.1 m/s (estimated from the ESROB measurements), which supports this assertion, demonstrating a high
CF region offshore of ESROB (Fig. 9a, d). Onshore advection of the high CF water originated from the
upwelling of nutrient rich water along the coast, accounting for half the CF change during the event (up to
1.60 μg/l d$^{-1}$ averaged over the E07 when $\partial$Chl $a/\partial t$ > 0) observed at ESROB during E07 (Fig. 7).
Conversely, offshore advection of high CF coastal plume water of northern origin may also be significant as
that of E10. Based on previous research conducted in other coastal systems, E10 is similar to results on
temporal and spatial variations of CDOM, CF, and primary productivity by cross-shore (onshore and/or
offshore) advection of high SST and high CF plume water associated with local circulations (Brzezinski and
Washburn, 2011;Warrick et al., 2007). Thus, cross-shore advection of low SSS and high CF water associated
with ambient circulation plays an equally significant role in shaping and triggering bloom events in the
coastal area.

### 4.2. Other mechanisms

The high CF events observed at ESROB are not local blooms triggered by either nutrients or light
availability. The upward vertical flux of nitrate into the euphotic zone at Huntington Beach, southern
California shows how vertical nutrient supply triggers local chlorophyll blooms (Omand et al., 2012). Omand
et al. (2012) demonstrated that each episodic bloom was preceded by a vertical nitrate flux event 6–10 days
earlier using nitrate concentrations estimated from a temperature proxy. Relationships between nitrate and
temperature and between nitrate and salinity observed from the surveys in July and October of 2011 and
April of 2012 are not significantly different each other, and the vertical nitrate fluxes were estimated by the
temperature proxy to discuss the potential role of nitrate in triggering the episodic blooms. However, both
advective and turbulent nitrate fluxes estimated using a nitrate proxy utilized from temperature
measurements (Fig. 10) did not account for the observed CF blooms (not shown). Although some episodic
CF blooms (E01 and E06) are preceded by flux peaks with a typical time lag of 4–12 d, most events are not
directly linked to the variability in vertical nitrate fluxes, suggesting only minor roles of nutrient flux in
shaping CF variability observed at ESROB in summer.
Time-series of the euphotic zone (Z$_{eu}$) was compared with others to examine the effects of light adaptation
on the bloom events from two PAR sensors available for 2012 and 2013 (Figs. 6, 7). Basically, Z$_{eu}$ of 18 m



averaged over E04–E10 was deeper than 10.5 m which is $Z_{eu}$ averaged over the two whole summer periods (JJAS), indicating that the light environment was favorable at least for retaining and increasing of the CF bloom observed at ESROB. $Z_{eu}$ of 20 m averaged over the three bloom events (E04–E06) in 2012 was deeper than that ($Z_{eu}$=15 m for E07–E10) in 2013, supporting more favorable CF bloom conditions in 2012 than 2013. Correspondingly, CF of 1.8 µg/l averaged over E04–E06 in 2012 was higher than that in 2013 (~ 1.6 µg/l for E07–E10). Our results on the deeper $Z_{eu}$ with higher CF in 2012 than 2013 summers are consistent with those in other systems (e.g., Mississippi River coastal system) where light attenuation plays a significant role in increasing phytoplankton biomass, and productivity variation (Lehrter et al., 2009). However, the CF changes among the individual events do not necessarily follow $Z_{eu}$ variations (Table 1), suggesting a minor role of light availability in shaping the CF variability observed at ESROB.

### 4.3. Inter-annual variations

The CF bloom events near the coast can vary inter-annually depending on the passage of typhoons. Five typhoons passed through this area were associated with the CF bloom events for two summers (2011 and 2012) and there was no typhoon affecting the CF bloom events in 2013 summer. Both strong wind forcing and intensive rainfalls associated with typhoon passage nearby determine how the plume water is advected in and around ESROB, which varies year-to-year. In 2011, for example, the CF enhancement (E01) was accompanied by the passage of MAON (equatorward wind stress and current) through the area south of ESROB, whereas E03 ended with the passage of MUIFA (poleward wind stress and current) passing through the area north of ESROB (Fig. 5b). Similarly, surface CF decreased (increased) with the passages of typhoons KHANUN and BOLAVEN (TENBIIN) through the area north (south) of ESROB (Fig. 6b). Without any typhoon passage in the summer of 2013, only half the CF events could be explained by the alongshore advection contrasting with those in the other two years (Fig. 7b). Thus, the primary productivity in the area is possibly affected severely by inter-annual variations of typhoon-induced alongshore advection.

Remote wind forcing significantly affecting summer-time equatorward currents near the coast via equatorward propagating coastal trapped waves (CTWs) varied in the summers of 2011, 2012, and 2013 (Park et al., 2018 submitted). The CTWs generated off the Russian coast (~1,000 km from ESROB) changed equatorward currents at the location of ESROB to yield more equatorward advection in 2011 and 2012 summers and more poleward advection in 2013 summer, of low-salinity plume water near the coast (Park et al., 2018 submitted). These results may be relevant to more CF bloom events explained by equatorward advection of plume water of northern origin in 2011 and 2012 summers than 2013 summer (6 among 6 events vs. 2 among 4 events). Therefore, inter-annual variations of alongshore advection and surface CF blooms near the coast are possibly affected the CTWs propagating equatorward from the Russian coast, where wind forcing varies considerably to generate CTWs. Park et al. (2018 submitted) also quantified the impact of EKWC on the alongshore current variability near the coast, which yields less EKWC impact and more equatorward currents near the coast in 2011 and 2013 summers, than 2012 summer. Although this is inconsistent with less CF bloom events explained by the equatorward advection of plume water of northern



origin in 2013 summer, cross-shore advections of high CF water of either northern (E10) or southern origin
(E07) are possibly associated with EKWC recirculation based on the patterns of surface geostrophic currents
(Fig. 9).

**5. Concluding remarks**
The low-salinity chlorophyll-rich water originating from the northern coast accompanying heavy rainfalls is
often advected equatorward along the coast in summer, resulting in high surface CF enhancements near the
mid-east coast of Korea. Alongshore advection of high CF waters is primarily responsible for most (80 %, 8
of 10) of the CF events, which confirms that the bloom events are possibly controlled by the advection of
low SSS and high CF plume water in summer. In contrast to the bloom events associated with alongshore
advection, the high surface CF observed during E07 is possibly explained by the onshore advection of high
CF water of southern origin advected by the poleward-flowing EKWC. Similarly, offshore advection of high
CF coastal plume water of northern origin may be significant, as in the case of E10. Therefore, the
equatorward and cross-shore advections of chlorophyll-rich plume water with decreasing SSS plays a
primary role in the high productivity near the east coast of Korea in summer. Summer-time CF near the coast
varies inter-annually as the horizontal advections vary significantly, inter-annually associated with typhoon
passages nearby, CTWs generated from the Russian coast, and influence of the EKWC, which should be
addressed with long time series data in future.

**Acknowledgements**
We thank Kyung-Il Chang for his helpful comments to improve the original version of this paper.
Geostationary ocean color satellite data (chlorophyll and total suspended sediment), precipitation (rain), and
satellite altimetry-derived sea surface height data were provided by the KIOST (Korea Institute Ocean
Science and Technology), KMA (Korea Meteorological Administration), and AVISO (Archiving, Validation,
and Interpretation of Satellite Oceanographic data), respectively. This research was supported by the Basic
Science Research Program through the National Research Foundation of Korea (NRF) funded by the
Ministry of Education (No. 2017R1D1A1B03035958 and NRF-2015R1D1A1A02062252), and was a part
of the project titled "Deep Water Circulation and Material Cycling in the East Sea", funded by the Ministry
of Oceans and Fisheries, Korea. This work is partly supported by the Agency for Defense Development
(UD170006DD), Korea.

**Data availability:** All data are available upon request to the authors.
**Author contributions:** This work was conceptualized and funding was secured by SHN and YTS. In-situ
measurements were designed by SHN and YTS, and performed by YTS and JHP with equipment provided



by SHN and YTS. Data were analyzed by YTS and JHP. The manuscript was written by YTS and SHN and

edited by YTS, SHN, and JHP. All authors have approved the final article.

**Competing interests:** The authors declare that they have no conflict of interest.

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





Table 1. Sea surface temperature (SST) in °C, sea surface salinity (SSS) in g/kg, chlorophyll-a fluorescence (CF)
in µg/l, duration in day, and euphotic depth ($Z_{eu}$) in m during the E01–E10 observed from the surface mooring


|  |  | SST | SSS | CF (start & end dates) | Duration | $Z_{eu}$ |
|---|---|---|---|---|---|---|
|  | E01 | 20.5 | 31.2 | 1.65 (21. Jul. ~ 25. Jul.) | 4.9 | Not available |
| 2011 | E02 | 22.3 | 30.9 | 1.91 (26. Jul. ~ 03. Aug.) | 8.3 | Not available |
|  | E03 | 24.3 | 29.9 | 1.61 (05. Aug. ~ 08. Aug.) | 2.5 | Not available |
|  | E04 | 21.4 | 32.9 | 1.67 (16. Jul. ~ 20. Jul.) | 3.5 | 22 |
| 2012 | E05 | 22.8 | 32.8 | 1.29 (21. Jul. ~ 27. Jul.) | 5.8 | 20.6 |
|  | E06 | 18.1 | 33.4 | 2.35 (29. Aug. ~ 05. Sep.) | 6.4 | 16.8 |
|  | E07 | 16.1 | 34.1 | 1.6 (01. Jul. ~ 04. Jul.) | 2.3 | 17.8 |
|  | E08 | 21.2 | 33.2 | 1.6 (12. Jul. ~ 16. Jul.) | 4.4 | 15.7 |
| 2013 | E09 | 25.0 | 32.1 | 1.7 (28. Jul. ~ 01. Aug.) | 4.3 | 12.7 |
|  | E10 | 26.7 | 31.9 | 1.4 (18. Aug. ~ 23. Aug.) | 5.9 | 15.2 |





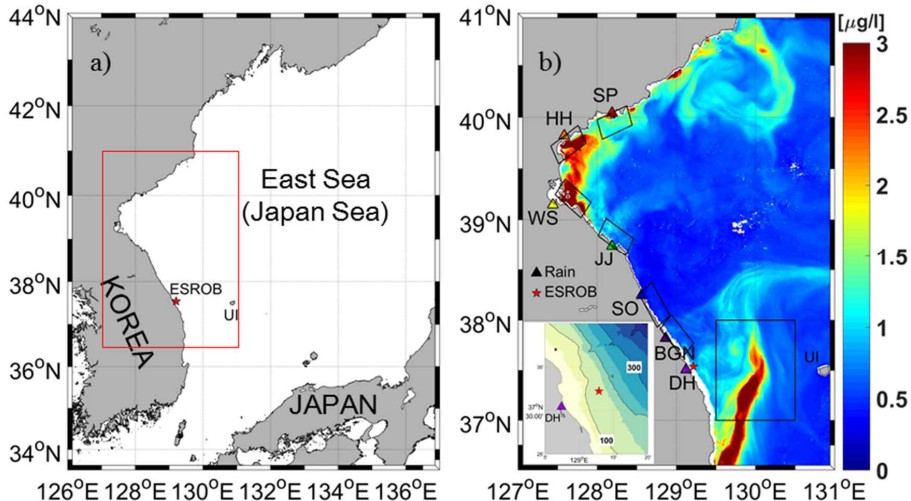

Figure 1. a) Study area in the western part of the East Sea (Japan Sea). b) A chlorophyll a image from the
geostationary ocean color satellite on September 6, 2012 in the area marked by red box in a). Black solid boxes
denote the areas where the chlorophyll a and TSS are averaged. Locations of the rainfall station along the east
coast of Korea are marked by triangles (SP: SinPho, HH: HamHeung, WS: WonSan, JJ: JangJun, SO: Sockcho,
BGN: BukGangNeung, DH: DongHae, rainbow colored). The surface mooring (ESROB) is indicated by a red
star in b) with bottom topography in the lower left corner where numbers denote water depth in meter (contour
interval: 100 m). Ulleung Island (UI) is located at ~ 131 °E.



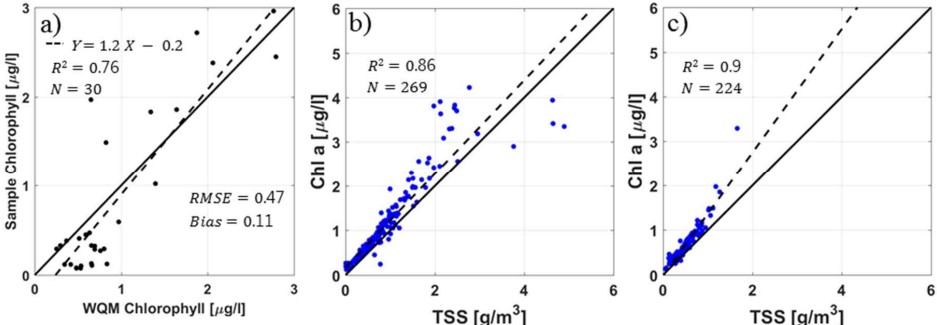


Figure 2. Results of cross-correlation ($R^2$: correlation coefficient) and linear regression analyses (dash lines)
between a) chlorophyll fluorescence measured by the ESROB WQM and absolute chlorophyll concentration
obtained from in-situ water samples; and between TSS and chlorophyll a concentration for b) the areas along and
near the east coast of Korea and c) area off the coast between DH and UI. The water samples (N: sample number)
were collected in July and October of 2011 and April of 2012.





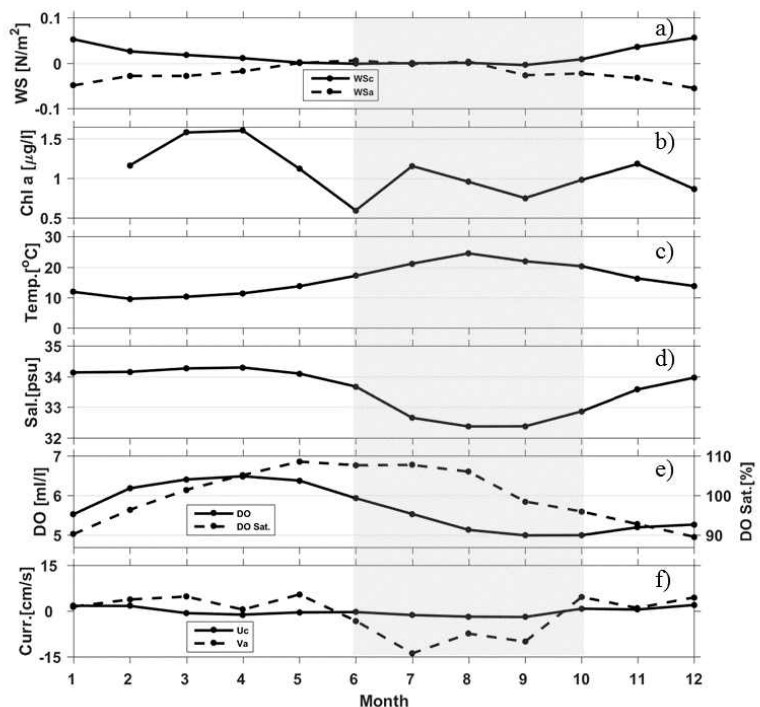


Figure 3. Climatology for a) alongshore and cross-shore components of wind stress, b) chlorophyll fluorescence,
c) water temperature, d) salinity, e) dissolved oxygen in both ml/l and percentage saturation, and f) alongshore
and cross-shore components of surface (~ 5 m) current constructed using ESROB data collected in three years
from 2011 to 2013. Summer season (JJAS) is shaded.






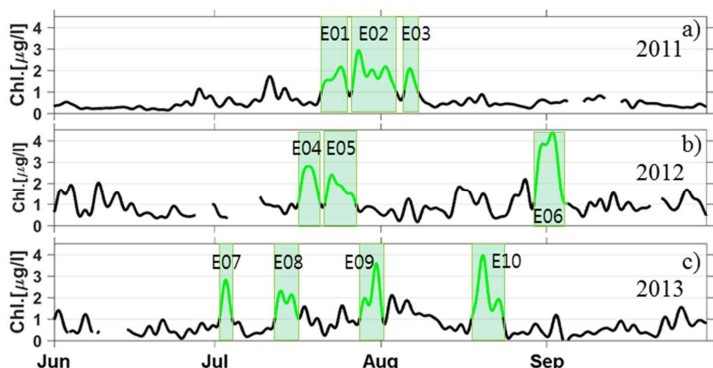


Figure 4. Time-series of low-pass filtered (cutoff period of 40 h) chlorophyll fluorescence observed at the ESROB
during the three summers (JJAS) of a) 2011, b) 2012, and c) 2013. The episodic bloom events are green-shaded
and labeled E01 to E10.



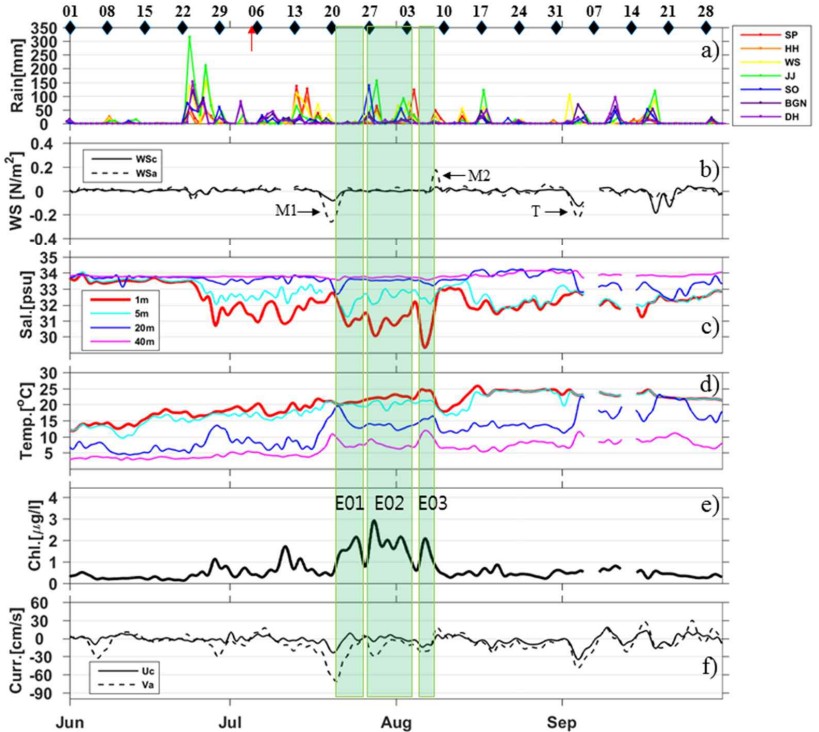


Figure 5. Time-series data collected in 2011 of a) daily rainfall amounts observed at weather stations (SP: SinPho,
HH: HamHeung, WS: WonSan, JJ: JangJun, SO: Sockcho, BGN: BukGangNeung, DH: DongHae) along the east
coast of Korea, and b) alongshore (solid) and cross-shore (dash) wind stresses, c) salinities, and d) water
temperatures observed at surface (red), 5 (cyan), 20 (blue), and 40 m (pink), e) surface CF, and f) alongshore
(dashed) and cross-shore (solid) currents, observed at the ESROB. The bloom events are labeled by E01 to E03.
In the top axis of (a), dates/times of satellite altimetry-derived surface geostrophic current map and geostationary
satellite ocean color image are remarked with black diamonds and red arrow, respectively. Nearby passages of
typhoons are indicated by black arrows in b) (M1: MAON, M2: MUIFA and T: TALAS).



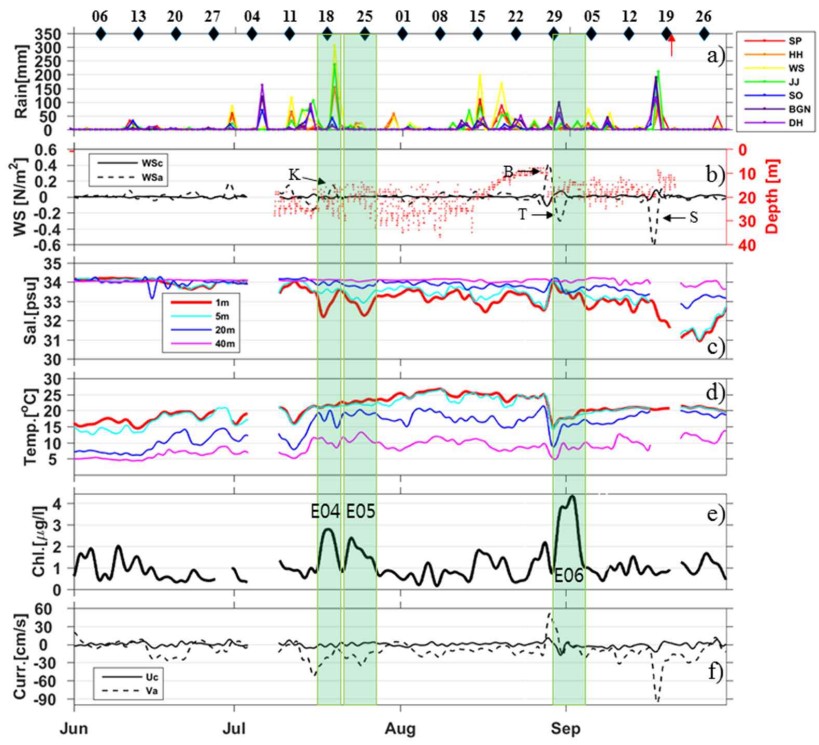


Figure 6. Same as Figure 5, except for 2012 bloom events labeled E04 to E06, and four typhoons (K: KHANUN,
T: TENBIN, B: BOLAVEN, S: SANBA). Euphotic depth ($Z_{eu}$, red dots) derived from two PAR sensors attached
to the ESROB are superimposed in b).






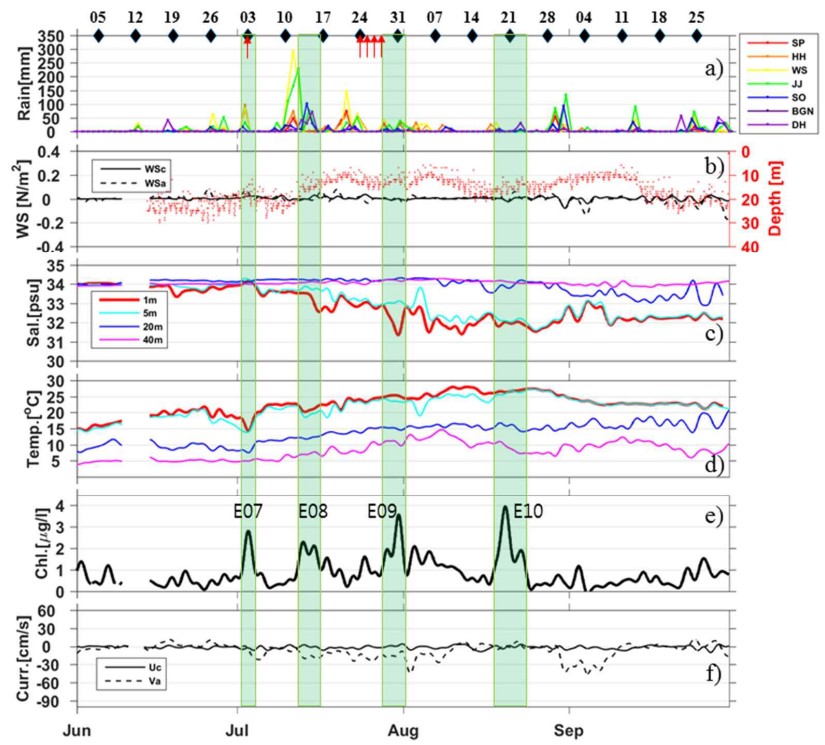


Figure 7. Same as Figure 6 except for 2013 bloom events labeled E07 to E10, and no typhoon occurrence.





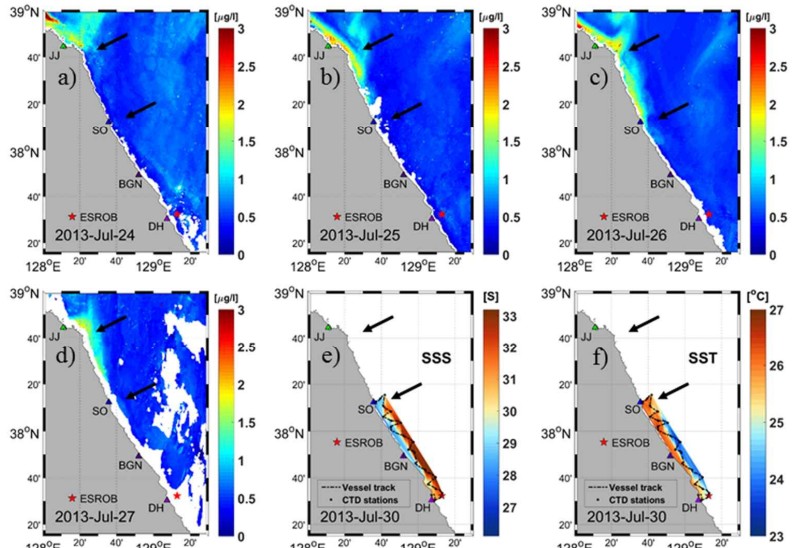

Figure 8. a)-d) Daily series of geostationary satellite ocean color images indicating surface chlorophyll a distributions from July 24 to 27, 2013. Surface distributions of e) salinity and f) temperature observed using a small research vessel (ship tracks and CTD stations are remarked with dashed lines and dots) in July 30, 2013 a couple of days after heavy rainfall in the region. Two black arrows in each panel head for the same locations in the vicinity of JJ (JangJun) and SO (Sockcho).





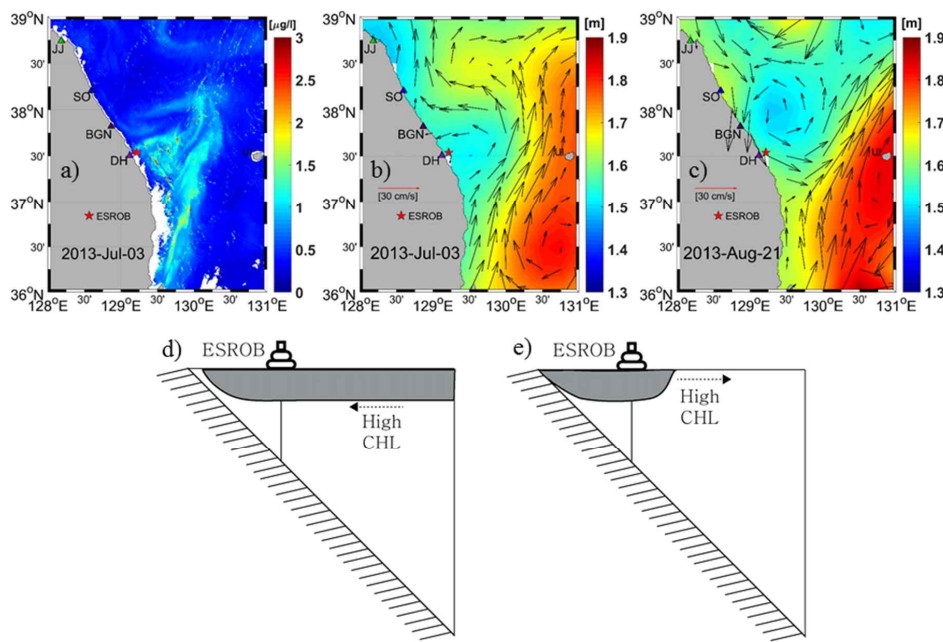

451

Figure 9. Distributions of a) daily composite of chlorophyll a concentration in July 3, 2013, obtained from the
geostationary satellite ocean color imager, and satellite altimetry-derived surface geostrophic currents in b) July
3 and c) August 21, 2013. Schematics for (d) on-shore and (e) off-shore advections of high CF surface water for
July 3 (E07) and August 21 (E10), 2013.






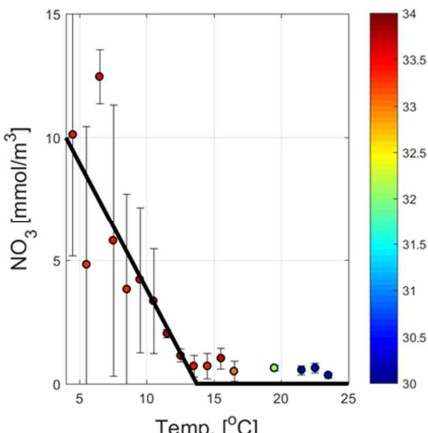


Figure 10. A linear fit (bold line) between temperature (Temp.) and nitrate ($NO_3$) for Temp < 14.0 °C ($NO_3$= 0 for
Temp > 14.0 °C) to observations near the east coast of Korea in summers of 2011 and 2012. A standard deviation
of nitrate and absolute salinity in g/kg are shown with vertical bars and colors (colorbar in the right), respectively.