# Peer review of "Summer-time episodic chlorophyll-a blooms near east coast of Korea"

_Biogeosciences, 2018_

## Referee Comment (RC1) · Anonymous Referee #1 · 4 Jun 2018

Review of:

Summer-time episodic chlorophyll-a blooms near east coast of Korea

by Young-Tae Son, Jae-Hyoung Park, and SungHyun Nam

Manuscript ID: BG 2018 183

General comments

This paper deals with summertime coastal phytoplankton blooms off eastern Korea, as measured at the ESROB fixed buoy site. The Authors show how advection of chl-rich, low salinity (due to typhoon-related heavy rainfall), water to the site triggers chl blooms, during which chl a reaches 4 ug/l and beyond. I find this paper very interesting, both

in a phenomenological sense and because of the completeness of the parameters measured at ESROB, accompanied by satellite imagery. The text is well written even though it needs a little English improvement. However, I have some reserves before recommending publication, which are explained in detail below in the particular comments. To sum them up, I am mainly concerned with: 1) the fact that Total Suspended Matter (TSS) and chl a don't co-vary for chl a > 3 ug/l, so high chl may actually be due to non-chl particulate optical signature; 2) the interpretation of the dynamic situation, i.e. I have a problem with the wind re-stratifying a water column. Also, upwelling is visible in the ESROB T and S record during poleward wind events, but it is not mentioned: even though it is not relevant for blooms, it should be, to make the physical interpretation complete 3) the lack of the description of the method for which the Authors find out that advection "... is primarily responsible for most (80 %) of the CF events" 4) the fact that salinity at ESROB never goes below 30 g/kg, except for E03, while SSS data indicate 27-29 g/kg for the plume in 2013. So does the plume really reach ESROB? Or maybe a mix of plume and offshore water? 5) the lack the suggestion for the mechanism for which the northern plume or southern chl-rich waters trigger blooms; also, why are there no other strong blooms north of ESROB? Has the nutrient load of such surface, fresh waters anything to do with bloom triggering?

Therefore, I do not recommend publication, but I do strongly encourage the Authors to revise the manuscript and challenge the above issues. Details can be found below, and the Authors can contact me via the Editor, anytime for any questions.

Form

The English of the manuscript is quite correct, but needs improvement. I tried to help with the list of corrections suggested below.

Particular comments and suggested text corrections

Title

Replace "summer-time" with "summertime" (hereafter "replace" will be represented as "->"). Please correct this also in the rest of the text.

"near east coast" -> "near the east coast"

Abstract

Line 23. Replace "accompanied" -> "were accompanied by".

1. Introduction

Line 37. "among others" -> "among other phenomena" Line 40. "plume-delivering" -> "plume- delivered". Line 42. "and significant" -> "and a significant". Line 43. "plume" -> "plumes". Line 44. "demonstrating localized"-> "demonstrating that localized". Line 45. What do you mean by "diversion"? Line 47. "differences with respect to the plume" -> Do you mean "differences between the plume and surrounding waters"? Please clarify. Line 57. "limited short-duration" -> " short-duration ". "limited" is redundant.

2 Data and Methods

Line 68. "The data collected includes" -> "The collected data include" Line 70-71. "vertical profile of current" -> "current vertical profiles" Line 71. "upper most" -> "upper-most".

Line 76. "alongshore current" -> "alongshore northward current", right? Line 79. "is needed to calibrate" -> "always needs calibration". Line 79. "owing to long-term sensor drift" -> "owing both to long-term sensor drift and to the fact that different chl a concentrations may yield the same fluorescence energy, i.e. the same number from the fluorimeter, because of temporal differences in phytoplankton species assemblage and

of the adaptation of species to different light conditions". I should add this because, as the Authors know, this point is very important if one wants to obtain realistic chl a quantities from a fluorimeter. See for example Longhurst et al., Prog. Oceanog. VoL 22, pp. 47 - 123, 1989, but also more recent references, with which I'm sure the Authors are familiar. Line 92. "geostationary ocean color satellite" -> is this the "NASA Geostationary Ocean Color Imager (GOCI)"? If not, which satellite? By the way, if you write the acronym of the satellite you can use it in the text instead of repeating "the ocean color satellite" every time. Line 94. "... at a grid 50 times further...". What do you mean? Please rephrase. I know that polar orbiting OC products have resolution of from 1 km onwards, so why 50 times? The 500 m grid would be 2 to 5 times finer, maybe. If "further" means "finer", that is. Line 95. "by the total" -> "by total". Line 98. "software modules applying a correction algorithm for the TSS and CDOM". Please cite software name and authors, as well as the reference or SW manual. These modules should be well described for anyone who might want to use them, because these corrections are very important. Line 101-102. "This indicated that Chl a can be measured regardless of the TSS both in the coastal and outer sea" -> I disagree: from Fig. 2b it seems that your Chl a measurements co-vary with TSS significantly only up to chl a = 3 ug l-1 and TSS = 2 mg m-3. This means that, up to these values, TSS is reasonably made only of phytoplankton. But when TSS is high, other particulate besides phytoplankton is present, so your satellite chl a algorithm may fail because it may mistake the light signal coming from other particulate for phytoplankton. Please comment or correct phrase. Is this issue crucial for what follows? That is, how much of the CF peaks in Fig. 4 is actually due to non-phytoplankton fluorescence? I say this because the peak values are beyond the range of chla-TSS tight covariance. So are they really phytoplankton blooms? Suggestion: why not over-plot the in situ chl a data in Fig. 4, e.g. as asterisks or crosses? This would make sure that the peaks are real chl a.

Line 106-107. "Precipitation in unit of mm/day recorded" -> "Precipitation (mm/day) was recorded". Line 109. "were proxied as freshwater..." -> I think there is a piece of

sentence missing.

3. Results

3.1. Climatological CF variations

Line 21. "trophic situation" -> "the trophic situation". Line 120. "upper most" -> "uppermost". Line 121. "summer-time" -> "summertime".

3.2. CF events observed in summers of 2011, 2012, and 2013

Line 134. "over considerable period (Fig. 4, Table 1)" -> "over a considerable period, i.e. days to weeks (Fig. 4, Table 1)". Line 136. "when CF > 1.0 $\mu$g/l" -> I don't understand this third condition, given the first two. Line 137. "three each year" -> "three in each year" Line 141. "rainfalls" -> "rainfall". For this word, usually plural not used. Pls correct also rest of manuscript.

Line 152. "wind stress, strong" -> "wind stress, the strong". Line 153. "developed before E04" -> should this not be "developed before and during most of E04"? Refer to Fig. 5. Line 157. Eliminate "both" if you use parentheses for opposing effects. Line 156-159. "poleward (equatorward) wind stress re-stratified (well-mixed)" -> In my opinion, it is impossible for wind stress to re-stratify a water column, no matter its direction. So, I think that poleward (and, by the way, strong) wind stress cannot re-stratify, but mix only. Indeed, if one looks at the T and S time series of Fig. 6c and d, in correspondence of the B poleward event, the isotherms drop and the isohalines rise, but remain separated, except for the 1 and 5 m isolines. This indicates upwelling, which is consistent with the wind and coast configuration. Next, during the T (equatorward) event the isotherms rise and the isohalines drop, indicating downwelling, and after the events the isolines settle to their normal values. So mixing is not so visible, to my opinion. However, Authors are right about mixing for the M1, T and S events, all equatorward, when

the 1, 5 and 20 m isolines join. In sum, it looks like during equatorward events the mixing takes place and stays there so it can be measured (Ekman transport is onshore). On the other hand, during poleward events, mixing probably takes place, but is either less intense or is not visible at ESROB, because mixed water is displaced offshore by Ekman transport, and only the "frictionless" effect of upwelling is measurable. What are the Authors' comments? Line 164 "did not accompany preceding heavy" -> "did not follow heavy".

**3.3. Surface CF distributions**

Line 176. "(Fig. 9a, b, c and d)" -> "(Fig. 8a-d)" Should this be Fig. 8, not 9? Line 177. "e.g. off the SP, HH, and WS," -> "e.g. off the SP, HH, and WS sites," Lines 178 - 179. "and extended" -> "while a more coastal branch extended" Line 179. "(Fig. 9a, b, c and d)" -> "(Fig. 8a-d)" Again should it be Fig. 8? Lines 179-180. "coast during the period (Fig. 9a, b, c and d) after the heavy rainfalls in July 19–24 (Fig. 7a)." -> "coast (Fig. 8a-d) after the heavy rainfalls of July 19–24 (Fig. 7a)."

**Page 7**

Line 184. "(Fig. 9e and 9f)" -> "(Fig. 8e, f)". Again Fig. 8. Line 188. "A pattern of" -> "The patterns of" Line 190. "within cyclonic" -> "within the cyclonic" Lines 194-195. "coastal zone" -> "coastal zone, near DH and ESROB, as well as equatorward currents just to the north". Line 196. "(as cases of many other events, see Fig. 1 or Fig. 8)" -> "(see Fig. 7, but also other similar events, as in Fig. 1 or Fig. 8)"

**4.1. Horizontal advection**

Line 203. "and is primarily responsible for most (80 %) of the CF events." Please tell how the Authors checked this, practically. Did they see if the plume could reach ESROB for each event, given the duration of the event and the equatorward current? Did they use the current at ESROB or available imagery, as in the example of Fig. 8? Line 205. "measured to 100 km (= dy)" -> "measured to be dy = 100 km" Line 206. "with Chl

a change of about 2.5 $\mu$g/l". Is this the difference between chlorophyll at the plume source and the initially oligotrophic water at ESROB? If not, between which points is this difference computed? Please specify. Line 207. "(Fig. 9a, b, c and d)." -> "(Fig. 8a-d)." Again Fig. 8. Lines 201-214. I understand the calculation and it is good that the computed advective rate of change matches local change at ESROB. However, I do notice that the maximum chl in the plume doesn't exceed 2.5 $\mu$g/l (Fig. 8), and that this is the source value at the plume's origin, which never moves south. Indeed, the water that eventually reaches ESROB has much lower chl, according to Fig. 8, i.e. max 1-1.5 $\mu$g/l. So how can E09 reach a peak of 3.5 $\mu$g/l (Fig. 7) if it is only fueled by the plume? Maybe the plume is more important as a nutrient carrier than a chl carrier, so arriving at ESROB it triggers a bloom? However, if so, why are there no other strong blooms north of ESROB in the images of Fig. 8, but only the southward-decreasing chl a plume signal? If the plume is the responsible for the blooms, then there should be even more intense blooms north of ESROB. Am I missing something? Please comment/revise in text. Also, concerning the plume investing ESROB: from ESROB and SSS cruise data in Figs. 7 and 8 one sees that S = 31.5 g/kg at ESROB at the peak of the E09 event, but the plume salinity seems much lower from Fig. 8e, i.e. S < 29. So which water reaches ESROB? It doesn't look like pure plume water; maybe it is a plume-offshore mix? I think the Authors should clarify this issue.

Line 224. "estimated to 0.86" -> "estimated to be 0.86" Lines 225-226. "demonstrating a high CF region offshore of ESROB (Fig. 9a, d)" -> "demonstrating the influence of the high CF region offshore on the ESROB site (Fig. 9a, d)". Do I understand well? Line 227. "nutrient rich" -> "nutrient-rich" Line 227. "accounting for half the CF change" -> "accounts for half the CF change". Question same as above: since E07's peak reaches 3.5 ug/l and the offshore water carries 1.6 ug/l, then where does the remaining chl (3.5 - 1.6 = 1.7 ug/l) come from? Line 230. "significant as that of E10" -> "significant, as happens for the E10 bloom". Do I understand well?

4.2. Other mechanisms

Line 244. "different each other" -> "different from each other" Line 251. "euphotic zone" -> "euphotic zone depth". Line 251. "was compared with others" - > "were compared with others". What do you mean by "others"? Line 252. "events from two PAR" -> "events, using two PAR" Line 252-3. "Basically, Zeu of 18 m averaged over E04–E10 was deeper" -> ""Basically, the average for the E04 to E10 bloom periods, Zeu = 18 m, was deeper".

Line 255. "Zeu of 20 m averaged" -> "A Zeu of 20 m obtained by averaging".

4.3. Inter-annual variations

Line 265. "typhoons passed through" -> "typhoons that passed through" Line 277. "summer-time" -> "summertime".

5. Concluding remarks

Line 296. "high surface CF enhancements" -> "high surface CF events" or "enhanced surface CF" Lines 297-299. "Alongshore advection... in summer". I think that this is my main concern about the paper. That is, the Authors have demonstrated that the blooms at ESROB are not driven by local vertical nutrient supply (text relative to Fig. 10). In addition, the Authors show that chl-rich plume waters or southern waters reach ESROB. So they argue that such advection is responsible for most events. But, I ask, how? What is the biogeochemical mechanism that triggers the blooms at ESROB, after the chl-rich water hits the site? This is not clearly stated. Especially since the advected waters that arrive at ESROB have only about half of the peak chl that is measured during blooms. Why is ESROB so special about blooms, with respect to the rest of the? Or maybe other blooms are visible in other sites? I suggest that Authors should (1) propose a mechanism for bloom generation (also tentative, that's OK); (2) discuss

the occurrence or lack thereof of such other blooms at other sites along the coast, by showing or commenting evidence from satellite imagery (or other available data). By the way: (3) is there an image showing any of the blooms at ESROB itself, to have an idea of the bloom's extension around the site?

Line 303. "the equatorward and cross-shore advections" -> "the equatorward and cross-shore advection". No need for plural.

Line 303. "SSS plays" -> "SSS play"

Tables, Figures and captions

Table 1 caption. "duration in day" ->"duration in days".

Line 405. Figure 1 caption. "water depth in meter" -> "water depth in meters" Line 422. Figure 4 caption. "at the ESROB" -> "at ESROB" Line 430. Figure 5 caption. "at surface" -> "at the surface" Line 431. Figure 5 caption. "at the ESROB" -> "at ESROB" Line 437. Figure 6 caption. "except for 2012 bloom events" -> "but for the 2012 bloom events" Line 442. Figure 7 caption. "except for 2013 bloom events" -> "but for the 2013 bloom events" Lines 446-7, Fig. 8 caption. "Surface distributions of e) salinity and f) temperature observed using a small research vessel (ship tracks and CTD stations are remarked with dashed lines and dots)" -> "In situ surface distributions of e) salinity and f) temperature (dashed lines: ship tracks; dots: CTD stations)" Line 459. Figure 10 caption. "in summers" -> "in the summers" Line 460. Figure 10 caption. "A standard deviation of" -> "Standard deviations for".

---

## Referee Comment (RC2) · Anonymous Referee #2 · 12 Jun 2018

The authors analyzed the summertime chlorophyll bloom in the East Sea (Japan Sea) based on the hydrographic data obtained at a fixed buoy site and satellite ocean color and sea surface height data. The subject matter of this manuscript is scientific interest to study the coastal chlorophyll a blooms in a short time scale. It is shown that horizontal advection is the key mechanism for the appearance of the summertime bloom events in the study area. However, more careful analysis is needed to accept it as the conclusion. I would suggest the publication of the manuscript after some moderate revisions, especially in interpretation of results and discussion.

I would like to point out the following comments that may help authors to improve their work.

Major comments:

[Figure]

Upwelling is frequently observed in the east coastal area in the northern hemisphere under the summertime monsoonal (poleward) wind. This means that upwelling can be a major contributor for the chlorophyll a blooming event. Enhancement of vertical mixing associated with the strong wind is also an important process for the local nutrient budget. In the beginning of event E06, temperature decreased in the whole water column which is due to the passage of typhoon as described in the text. However, the authors did not mentioned it as a possible governing mechanism for the blooming event. The documents by the National Institute of Fisheries Science of Korea show that July 2, July 11 and July 23-29, 2013 were the period of low temperature warning in the east coast of Korea including the study area. These periods are coincident with the blooming events. Thus, it must be carefully re-analyzed for the driving mechanism by using all available data, though no clear evidence for upwelling phenomena is shown in temperature data except E07 event in Figure 7. It would be better to show analytically whether the ESROB buoy site, i.e., the distance from the coast, is suitable to monitor the summertime coastal upwelling event.

Minor comments:

Line 76. Please provide the general width of the alongshore current, if possible.

Line 105. Please provide the source for precipitation data.

Line 144. '$\sim$ inducing strong equatorward (before E01)'. Both salinity and temperature increased sharply, especially in the lower layers just before E01 in Figure 5. This is not consistent with the effect of the equatorward flow.

Line 153. ' equatorward currents developed before E04'. Temperature increased in the whole water column before E04 under the equatorward current as well as before E01.

Line 176. 'a high surface CF zone in the northern area'. Is it a general feature in summer only? Why CF is high in the northern area?

Line 237. '4.2. Other mechanisms'. It would be better to add more discussion.

[Figure]

Lines 275, 281, 286. Check 'Park et al., 2018'. Park and Nam, 2018 ?

Line 297, '80%, 8 of 10'. It may not be conclusive.

---

## Author Comment (AC1) · 5 Jul 2018

Reviewer #1 General comments This paper deals with summertime coastal phytoplankton blooms off eastern Korea, as measured at the ESROB fixed buoy site. The Authors show how advection of chl-rich, low salinity (due to typhoon-related heavy rainfall), water to the site triggers chl blooms, during which chl a reaches 4 ug/l and beyond. I find this paper very interesting, both in a phenomenological sense and because of the completeness of the parameters measured at ESROB, accompanied by satellite imagery. The text is well written even though it needs a little English improvement. However, I have some reserves before recommending publication, which are explained in detail below in the particular comments. Thank you very much for the valuable comments below. We deeply appreciate the detailed suggestions and have

revised the manuscript based on these comments.

To sum them up, I am mainly concerned with: 1) the fact that Total Suspended Matter (TSS) and chl a don't co-vary for chl a > 3 ug/l, so high chl may actually be due to non-chl particulate optical signature The GOCI Chl a can be overestimated when the TSS is high as has also been shown in previous research. In the revised manuscript, we clarified this point, citing a new reference as below.

"Despite the fact that absolute value of Chl a can be overestimated at high TSS (Kim et al., 2016), this indicates . . ."

However, the relative value of GOCI Chl a in this area is still useful for understanding spatial Chl a distributions and their temporal variations.

Kim, W., Moon, J. -E., Park, Y. -J., and Ishizaka, J., Evaluation of chlorophyll retrievals from Geostationary Ocean Color Imager (GOCI) for the North-East Asian region, Rem. Sens. Environ., 184, 428–495, 2016.

2) the interpretation of the dynamic situation, i.e. I have a problem with the wind re-stratifying a water column. Also, upwelling is visible in the ESROB T and S record during poleward wind events, but it is not mentioned: even though it is not relevant for blooms, it should be, to make the physical interpretation complete We agree that the description in the original manuscript was not sufficient and could cause unnecessary confusion on the interpretation of mixing and re-stratifying dynamics. In general, i.e., with no coastal boundary, winds enhance mixing and only break stratification. However, the winds either increase or decrease the stratification near the coast owing to coastal up- and downwelling responses (with offshore and onshore Ekman transport in the upper layer) to alongshore wind, depending on its direction. The water column in the coastal area can be either re-stratified (downwelling favorable wind, Fig. R1 left) or ho-mogenized (upwelling favorable wind, Fig. R1 right) depending on the alongshore wind. Here, we believe the mixing process is minor compared to the upwelling/downwelling response with Ekman transport.

Figure R1. Schematics of isopycnals or isotherms (dashed line) and alongshore currents at the upper layer in response to downwelling (left) and upwelling (right) favorable wind stress.

To clarify this point, we revised the sentences in Section 3.2 as below.

"... and implying the downwelling (before E01) and upwelling (after E03) in the vicinity of ESROB."

"Since typhoon KHANUN drove poleward wind stress, the strong equatorward currents (showing downwelling induced by equatorward wind stress) developed before and during the most of E04 were weakened, and SSS increased to ..."

"Two typhoons (BOLAVEN and TENBIIN) successively passed the area and poleward (equatorward) wind stress imposed by BOLAVEN (TENBIIN) induced an upwelling (downwelling) response with poleward (equatorward) and offshore (onshore) transports at the upper layer, decreasing (increasing) water temperature, and increasing (decreasing) salinity in the whole column during E06."

3) the lack of the description of the method for which the Authors find out that advection "... is primarily responsible for most (80 %) of the CF events" We simply counted the number of events where the alongshore advection plays a primary role in changing the CF. More specifically, equatorward currents and salinity decreases were accompanied during the 8 events (E01–E06, E08–E09). The high CFs during the remaining two events (E07 and E10) were discussed in association with cross-advections. In the revised manuscript, we inserted supplementary information as "... primarily responsible for most (80%, 8 of 10) of the CF events."

4) the fact that salinity at ESROB never goes below 30 g/kg, except for E03, while SSS data indicate 27-29 g/kg for the plume in 2013. So does the plume really reach ESROB? Or maybe a mix of plume and offshore water? Great point. We believe that what was observed at the ESROB is not pure plume water, but water mixed with offshore water. The plume water salinity increases as mixed with the saline offshore water and the modified plume water is advected equatorward to the ESROB. In the revised manuscript, we clarified the point by inserting the new sentence below in Section 4.1.

"The distribution and temporal evolution of SSS observed in July 30, 2013 implies the low salinity plume water (SSS < 29 found in the northern coastal area, Fig. 8e) is mixed with saline offshore water while advected equatorward, yielding slightly higher (> 31) SSS at ESROB."

5) the lack the suggestion for the mechanism for which the northern plume or southern chl-rich waters trigger blooms; also, why are there no other strong blooms north of ESROB? Has the nutrient load of such surface, fresh waters anything to do with bloom triggering? As mentioned in 1), the GOCI CF can be overestimated and may not be comparable to the in-situ CF observed at the ESROB directly. We believe the CF is higher in the northern plume water than the modified water found at the ESROB; the CF maintained or decreased while advected equatorward. The opposite case (CF increased while advected) was tested by examining the possibility of local blooms triggered by either nutrients or light availability, but not supported as discussed in Section 4.2.

Replace "summer-time" with "summertime" (hereafter "replace" will be represented as "->"). Please correct this also in the rest of the text. This has been revised throughout the manuscript.

"near east coast" -> "near the east coast" This has been corrected.

Line 23. Replace "accompanied" -> "were accompanied by". This has been corrected.

Line 37. "among others" -> "among other phenomena" Line 40. "plume-delivering" -> "plume- delivered". Line 42. "and significant" -> "and a significant". Line 43. "plume" -> "plumes". Line 44. "demonstrating localized"-> "demonstrating that localized". These have been revised.

Line 45. What do you mean by "diversion"? We meant the change of dominant plankton species within the plume water by the "diversion." In the revised manuscript, we changed the sentence to "a few days after the plume water discharge" to avoid the confusion.

Line 47. "differences with respect to the plume" -> Do you mean "differences between the plume and surrounding waters"? Please clarify No, it means differences among the local plumes. To clarify this, the sentence was revised to "... revealing large Chl a differences among the local plumes."

Line 57. "limited short-duration" -> " short-duration ". "limited" is redundant. This has been corrected.

Line 68. "The data collected includes" -> "The collected data include" Line 70-71. "vertical profile of current" -> "current vertical profiles" Line 71. "upper most" -> "uppermost". This has been revised as recommended.

Line 76. "alongshore current" -> "alongshore northward current", right? This has been revised to "poleward alongshore current" to make it consistent with other expressions.

Line 79. "is needed to calibrate" -> "always needs calibration". This has been revised as recommended.

Line 79. "owing to long-term sensor drift" -> "owing both to long-term sensor drift and to the fact that different chl a concentrations may yield the same fluorescence energy, i.e. the same number from the fluorimeter, because of temporal differences in phytoplankton species assemblage and of the adaptation of species to different light conditions". I should add this because, as the Authors know, this point is very important if one wants to obtain realistic chl a quantities from a fluorimeter. See for example Longhurst et al., Prog. Oceanog. VoL 22, pp. 47 - 123, 1989, but also more recent references, with which I'm sure the Authors are familiar. Thank you for the reference. We agreed that the issues of temporal differences in species assemblage and species

adaptation to different light conditions are important to obtain realistic Chl a. In the revised manuscript, we included the point as commented by citing Longhurst et al. (1989).

Line 92. "geostationary ocean color satellite" -> is this the "NASA Geostationary Ocean Color Imager (GOCI)"? If not, which satellite? By the way, if you write the acronym of the satellite you can use it in the text instead of repeating "the ocean color satellite" every time. The GOCI is the first geostationary orbit satellite image sensor to observe an ocean color around the Korean peninsula, loaded on the Communication, Ocean, and Meteorological Satellite (COMS, launched in 2010) of South Korea. The data may also be distributed via NASA. We revised the text to use GOCI throughout the manuscript.

Line 94. "... at a grid 50 times further...". What do you mean? Please rephrase. I know that polar orbiting OC products have resolution of from 1 km onwards, so why 50 times? The 500 m grid would be 2 to 5 times finer, maybe. If "further" means "finer", that is. The geostationary satellite was positioned 50 times higher than low altitude polar orbit ones. We corrected the sentence to avoid the confusion.

Line 95. "by the total" -> "by total". This has been corrected.

Line 98. "software modules applying a correction algorithm for the TSS and CDOM". Please cite software name and authors, as well as the reference or SW manual. These modules should be well described for anyone who might want to use them, because these corrections are very important. The software and references providing detailed descriptions were inserted into the sentence.

Line 101-102. "This indicated that Chl a can be measured regardless of the TSS both in the coastal and outer sea" -> I disagree: from Fig. 2b it seems that your Chl a measurements co-vary with TSS significantly only up to chl a = 3 ug l-1 and TSS = 2 mg m-3. This means that, up to these values, TSS is reasonably made only of phytoplankton. But when TSS is high, other particulate besides phytoplankton is

present, so your satellite chl a algorithm may fail because it may mistake the light signal coming from other particulate for phytoplankton. Please comment or correct phrase. Is this issue crucial for what follows? That is, how much of the CF peaks in Fig. 4 is actually due to non-phytoplankton fluorescence? I say this because the peak values are beyond the range of chla-TSS tight covariance. So are they really phytoplankton blooms? Suggestion: why not over-plot the in situ chl a data in Fig. 4, e.g. as asterisks or crosses? This would make sure that the peaks are real chl a. As described in 1), the GOCI Chl a can be overestimated when the TSS is high as has also been shown in previous works. However, the overestimation issue is only for GOCI CF (Fig. 2b and 2c) and not for the ESROB WQM (Fig. 2a). Note that Fig. 4 does not show the GOCI CF but the ESROB WQM. We agree that the GOCI Chl a can be somehow affected by non-phytoplankton fluorescence when the TSS is high. This is why we limit our interpretation on the GOCI CF to relative (not absolute) values, which is still useful for understanding its spatial distributions and temporal changes although it is not directly comparable to the ESROB WQM data shown in Fig. 4. We tried to over-plot the in situ CF in Fig. 4 but decided not to replace the original as in-situ water samples were taken in very limited times having relatively wide CF ranges, which were not very useful for addressing the point above. Instead, we clarified the GOCI CF and ESROB WQM CF against the in-situ water samples in the revised manuscript.

Line 106-107. "Precipitation in unit of mm/day recorded" -> "Precipitation (mm/day) was recorded". This was revised as recommended.

Line 109. "were proxied as freshwater..." -> I think there is a piece of sentence missing. We inserted the missing part on the sentence.

Line 21. "trophic situation" -> "the trophic situation". Line 120. "upper most" -> "uppermost". We could not find "trophic situation." The word "upper most" was revised to "uppermost."

Line 121. "summer-time" -> "summertime". This was revised as recommended.
Line 134. "over considerable period (Fig. 4, Table 1)" -> "over a considerable period, i.e. days to weeks (Fig. 4, Table 1)". This was revised as recommended.

Line 136. "when CF > 1.0 _g/l" -> I don't understand this third condition, given the first two. In the revised manuscript, we clarified the conditions to define the CF events as the original sentence was confusing. The event was basically defined as a period when CF > 1.0 $\mu$g/l. To avoid selecting too many temporal fluctuations as events, we use a constraint to select the 10 events using additional criterion where the duration of CF > 2.0 $\mu$g/l was longer than 1 d.

Line 137. "three each year" -> "three in each year" Line 141. "rainfalls" -> "rainfall". For this word, usually plural not used. Pls correct also rest of manuscript. This was revised throughout the manuscript.

Line 152. "wind stress, strong" -> "wind stress, the strong". This has been revised.

Line 153. "developed before E04" -> should this not be "developed before and during most of E04"? Refer to Fig. 5. This is correct, the sentence has been revised.

Line 157. Eliminate "both" if you use parentheses for opposing effects. This has been revised.

Line 156-159. "poleward (equatorward) wind stress re-stratified (well-mixed)" -> In my opinion, it is impossible for wind stress to re-stratify a water column, no matter its direction. So, I think that poleward (and, by the way, strong) wind stress cannot re-stratify, but mix only. Indeed, if one looks at the T and S time series of Fig. 6c and d, in correspondence of the B poleward event, the isotherms drop and the isohalines rise, but remain separated, except for the 1 and 5 m isolines. This indicates upwelling, which is consistent with the wind and coast configuration. Next, during the T (equatorward) event the isotherms rise and the isohalines drop, indicating downwelling, and after the events the isolines settle to their normal values. So mixing is not so visible, to my opinion. However, Authors are right about mixing for the M1, T and S events,

all equatorward, when the 1, 5 and 20 m isolines join. In sum, it looks like during equatorward events the mixing takes place and stays there so it can be measured (Ekman transport is onshore). On the other hand, during poleward events, mixing probably takes place, but is either less intense or is not visible at ESROB, because mixed water is displaced offshore by Ekman transport, and only the "frictionless" effect of upwelling is measurable. What are the Authors' comments? We mostly agree with you, and as mentioned in 2), believe that the water column at the ESROB is re-stratified and homogenized in response to downwelling and upwelling favorable winds (Fig. R1 left vs. right), assuming the mixing process plays only a minor role compared to the upwelling/downwelling response with Ekman transport.

Line 164 "did not accompany preceding heavy" -> "did not follow heavy". This was revised as recommended.

Line 176. "(Fig. 9a, b, c and d)" -> "(Fig. 8a-d)" Should this be Fig. 8, not 9? Correct. This has been revised to Fig. 8.

Line 177. "e.g. off the SP, HH, and WS," -> "e.g. off the SP, HH, and WS sites," Lines 178 - 179. "and extended" -> "while a more coastal branch extended" Line This has been revised as recommended.

179. "(Fig. 9a, b, c and d)" -> "(Fig. 8a-d)" Again should it be Fig. 8? Correct. This has been revised to Fig. 8.

Lines 179-180. "coast during the period (Fig. 9a, b, c and d) after the heavy rainfalls in July 19–24 (Fig. 7a)." -> "coast (Fig. 8a-d) after the heavy rainfalls of July 19–24 (Fig. 7a)." This has been revised as recommended.

Line 184. "(Fig. 9e and 9f)" -> "(Fig. 8e, f)". Again Fig. 8. This has been revised.

Line 188. "A pattern of" -> "The patterns of" Line 190. "within cyclonic" -> "within the cyclonic" Lines 194-195. "coastal zone" -> "coastal zone, near DH and ESROB, as well as equatorward currents just to the north". Line 196. "(as cases of many other events,

see Fig. 1 or Fig. 8)" -> "(see Fig. 7, but also other similar events, as in Fig. 1 or Fig. 8)" This has been revised as recommended.

Line 203. "and is primarily responsible for most (80 %) of the CF events." Please tell how the Authors checked this, practically. Did they see if the plume could reach ESROB for each event, given the duration of the event and the equatorward current? Did they use the current at ESROB or available imagery, as in the example of Fig. 8? As mentioned in 3), we simply counted the number of events where the equatorward currents and salinity decreases were accompanied with the CF events. The exceptions are only two events (E07 and E10) where the equatorward currents and salinity decreases are not clear just before and during the event periods (Fig. 7).

Line 205. "measured to 100 km (= dy)" -> "measured to be dy = 100 km" This has been revised as recommended.

Line 206. "with Chl a change of about 2.5 _g/l". Is this the difference between chlorophyll at the plume source and the initially oligotrophic water at ESROB? If not, between which points is this difference computed? Please specify. Yes, it is the difference between CF at the plume source and initially oligotrophic water at ESROB as specified in the revised manuscript.

Line 207. "(Fig. 9a, b, c and d)." -> "(Fig. 8a-d)." Again Fig. 8. Correct. This has been revised to Fig. 8.

Lines 201-214. I understand the calculation and it is good that the computed advective rate of change matches local change at ESROB. However, I do notice that the maximum chl in the plume doesn't exceed 2.5 _g/l (Fig. 8), and that this is the source value at the plume's origin, which never moves south. Indeed, the water that eventually reaches ESROB has much lower chl, according to Fig. 8, i.e. max 1-1.5 _g/l. So how can E09 reach a peak of 3.5 _g/l (Fig. 7) if it is only fueled by the plume? Maybe the plume is more important as a nutrient carrier than a chl carrier, so arriving at ESROB it triggers a bloom? However, if so, why are there no other strong blooms north of

ESROB in the images of Fig. 8, but only the southward-decreasing chl a plume signal? If the plume is the responsible for the blooms, then there should be even more intense blooms north of ESROB. Am I missing something? Please comment/revise in text. Also, concerning the plume investing ESROB: from ESROB and SSS cruise data in Figs. 7 and 8 one sees that S = 31.5 g/kg at ESROB at the peak of the E09 event, but the plume salinity seems much lower from Fig. 8e, i.e. S < 29. So which water reaches ESROB? It doesn't look like pure plume water; maybe it is a plume-offshore mix? I think the Authors should clarify this issue. Great point. We tested the possibility of a local bloom triggered by nutrients advected equatorward and discussed in addition to that triggered by vertical nutrient supply in Section 4.2. However, the nutrient loading mechanisms were found to play only a minor role. Here, we have two issues using the GOCI CF data. One is that the GOCI CF cannot only overestimate but can also underestimate the Chl a when the TSS is high in the coastal area (Fig. 2b). We would limit our interpretation on the GOCI CF to relative (not absolute) values, not directly comparable to the ESROB WQM data shown in Fig. 4. Next is that the GOCI CF, unfortunately, is not available very near the coastal zone (see the blanks shown in white in Fig. 8a–8d). We believe the plume water having high (> 2.5 $\mu$g/l in GOCI CF scale) CF and low (< 29 g/kg) salinity advected equatorward very near the coast as in the wedge patterns of SSS and SST observed in July 30 (Fig. 8e and 8f) to reach the ESROB although slightly mixed by saline and low CF offshore water. The CF and SSS at ESROB would be $\sim$2.0 $\mu$g/l in absolute ESROB WQM scale and $\sim$32 g/kg, respectively. More quantitatively, we showed that the rate of CF change observed at the ESROB is comparable with that owing to equatorward advection (v times dChl/dy) in Section 4.1.

Line 224. "estimated to 0.86" -> "estimated to be 0.86" Correct. This has been revised to Fig. 8.

Lines 225-226. "demonstrating a high CF region offshore of ESROB (Fig. 9a, d)" -> "demonstrating the influence of the high CF region offshore on the ESROB site (Fig.

9a, d)". Do I understand well? Correct. This has been revised as recommended.

Line 227. "nutrient rich" -> "nutrient-rich" This has been revised as recommended.

Line 227. "accounting for half the CF change" -> "accounts for half the CF change". This has been revised as recommended.

Line 227. Question same as above: since E07's peak reaches $\sim$ (3.5 - 1.6 = 1.7 ug/l) come from? Note that the 1.6 $\mu$g/l/d is not CF itself in a unit of $\mu$g/l but the time rate of its change at the ESROB site in a unit of $\mu$g/l/d, i.e., CF change in a day. We clarified that the time rate of CF change (up to 1.6 $\mu$g/l/d averaged over the E07 for the period when $\partial$Chl a/$\partial$t > 0) and the contribution of cross-shore advection (0.86 $\mu$g/l/d) are comparable.

Line 230. "significant as that of E10" -> "significant, as happens for the E10 bloom". Do I understand well? Correct. This has been revised as recommended.

Line 244. "different each other" -> "different from each other" Line 251. "euphotic zone" -> "euphotic zone depth". This has been revised as recommended.

Line 251. "was compared with others" - > "were compared with others". What do you mean by "others"? We specified the word as "the other time-series data recorded at ESROB" in the revised manuscript.

Line 252. "events from two PAR" -> "events, using two PAR" Line 252-3. "Basically, Zeu of 18 m averaged over E04–E10 was deeper" -> ""Basically, the average for the E04 to E10 bloom periods, Zeu = 18 m, was deeper". This has been revised to "events, using the data collected with two PAR sensors . . ." The latter was revised as recommended.

Line 255. "Zeu of 20 m averaged" -> "A Zeu of 20 m obtained by averaging". Line 265. "typhoons passed through" -> "typhoons that passed through" Line 277. "summertime" -> "summertime". Line 296. "high surface CF enhancements" -> "high surface CF events" or "enhanced surface CF" This was revised as recommended.

Lines 297-299. "Alongshore advection... in summer". I think that this is my main concern about the paper. That is, the Authors have demonstrated that the blooms at ESROB are not driven by local vertical nutrient supply (text relative to Fig. 10). In addition, the Authors show that chl-rich plume waters or southern waters reach ESROB. So they argue that such advection is responsible for most events. But, I ask, how? What is the biogeochemical mechanism that triggers the blooms at ESROB, after the chl-rich water hits the site? This is not clearly stated. Especially since the advected waters that arrive at ESROB have only about half of the peak chl that is measured during blooms. Why is ESROB so special about blooms, with respect to the rest of the? Or maybe other blooms are visible in other sites? We think there was a misunderstanding possibly owing to the poor presentation of the original manuscript. Our conclusion is that there was no local bloom at ESROB, but the water having high CF was transported past the ESROB in the alongshore (equatorward) or cross-shore directions. The summertime equatorward current near the coast was the primary process accounting for the CF variability at ESROB.

I suggest that Authors should (1) propose a mechanism for bloom generation (also tentative, that's OK); There might be potential biogeochemical mechanisms that trigger local blooms at or nearby the ESROB. However, as discussed in Section 4.2, both nutrient loading and changing light availability at ESROB hardly accounted for the observed CF variability. Based on our results, we believe that the blooms triggered in remote places (particularly in the northern coastal area) by some mechanisms, such as advective and diffusive nutrient supplies from rivers/rainfall or changing euphotic depth, and the CF-rich water are frequently transported (particularly equatorward) into the ESROB site in summer.

(2) discuss the occurrence or lack thereof of such other blooms at other sites along the coast, by showing or commenting evidence from satellite imagery (or other available data). The GOCI CF, although the absolute values are not very useful, often shows higher CF in the northern coastal areas in summer along the east coast of North Korea where the in-situ data are not available. A typical GOCI CF image such as Fig. 1 supports such blooms occurring more frequently in the northern than southern coastal areas in summer. However, there is another source of CF-rich water originating from the southern coastal area, which may be related to frequent summertime coastal upwelling off the southeastern coast of Korea. The ESROB is located in the area affected by both sources of high CFs although the equatorward advection prevailed at and inshore of the site in the three summers.

By the way: (3) is there an image showing any of the blooms at ESROB itself, to have an idea of the bloom's extension around the site? We hoped to find high CF at ESROB from the GOCI images for all the event and non-event periods in the three summers. However, we could not find this, as the GOCI CF data are not available very near the coast (blank area near the coast shown with white color in Fig. 8d) including the ESROB site owing to cloud cover.

Line 303. "the equatorward and cross-shore advections" -> "the equatorward and cross-shore advection". No need for plural. Line 303. "SSS plays" -> "SSS play" This was revised as recommended.

Table 1 caption. "duration in day" ->"duration in days". Line 405. Figure 1 caption. "water depth in meter" -> "water depth in meters" Line 422. Figure 4 caption. "at the ESROB" -> "at ESROB" Line 430. Figure 5 caption. "at surface" -> "at the surface" Line 431. Figure 5 caption. "at the ESROB" -> "at ESROB" Line 437. Figure 6 caption. "except for 2012 bloom events" -> "but for the 2012 bloom events" Line 442. Figure 7 caption. "except for 2013 bloom events" -> "but for the 2013 bloom events" This was revised as recommended.

Lines 446-7, Fig. 8 caption. "Surface distributions of e) salinity and f) temperature observed using a small research vessel (ship tracks and CTD stations are remarked with dashed lines and dots)" -> "In situ surface distributions of e) salinity and f) temperature (dashed lines: ship tracks; dots: CTD stations)" This was revised to "Surface

distributions of in-situ e) salinity and f) temperature (dashed lines: ship tracks; dots: CTD stations) ..."

Line 459. Figure 10 caption. "in summers" -> "in the summers" Line 460. Figure 10 caption. "A standard deviation of" -> "Standard deviations for". This was revised as recommended.

Please also note the supplement to this comment:
https://www.biogeosciences-discuss.net/bg-2018-183/bg-2018-183-AC1-supplement.pdf

---

## Author Comment (AC2) · 5 Jul 2018

Reviewer #2 In major comments, 1) Upwelling is frequently observed in the east coastal area in the northern hemisphere under the summertime monsoonal (poleward) wind. This means that upwelling can be a major contributor for the chlorophyll a blooming event. Enhancement of vertical mixing associated with the strong wind is also an important process for the local nutrient budget. In the beginning of event E06, temperature decreased in the whole water column which is due to the passage of typhoon as described in the text. However, the authors did not mentioned it as a possible governing mechanism for the blooming event. Thank you very much for the constructive comments. We agree that the coastal upwelling off the east coast in the northern hemisphere under the summertime monsoonal wind and intermittently enhanced vertical mixing associated with strong wind are a major driver to trigger the bloom via nutrient supply to the euphotic zone in general. In the revised manuscript, we explicitly mentioned the up- and downwelling responses to the strong alongshore wind associated with the typhoon passages. However, as discussed in Section 4.2, local blooms triggered by nutrient loading may play a minor role here in shaping the CF variability/events at ESROB.

2) The documents by the National Institute of Fisheries Science of Korea show that July 2, July 11 and July 23-29, 2013 were the period of low temperature warning in the east coast of Korea including the study area. These periods are coincident with the blooming events. Thus, it must be carefully re-analyzed for the driving mechanism by using all available data, though no clear evidence for upwelling phenomena is shown in temperature data except E07 event in Figure 7. It would be better to show analytically whether the ESROB buoy site, i.e., the distance from the coast, is suitable to monitor the summertime coastal upwelling event. Good point. Thank you for the information. Yes, we agree that the low temperature warning in July 2013 is relevant to the coastal upwelling off the east coast of Korea. The ESROB is well located where both up- and downwelling responses to local wind can be frequently monitored as has been known for decades (most recent reference is Park and Nam [2018]). However, please note that most CF events observed in the three summers are not directly linked to the nutrient fluxes enhanced by upwelling, but the equatorward advection of CF-rich plume water in the northern coastal area.

In minor comments Line 76. Please provide the general width of the alongshore current, if possible. The general width was inserted in the revised manuscript as recommended.

Line 105. Please provide the source for precipitation data. The data source was inserted in the revised manuscript as recommended.

Line 144. '∼ inducing strong equatorward (before E01)'. Both salinity and tempera-

[Figure]

ture increased sharply, especially in the lower layers just before E01 in Figure 5. This is not consistent with the effect of the equatorward flow. Line 153. ' equatorward currents developed before E04'. Temperature increased in the whole water column before E04 under the equatorward current as well as before E01. We understand the original manuscript may cause unnecessary confusion with this interpretation. As commented by this and another reviewer, we included the up- and downwelling responses as schematically shown in Figure R1 below. The winds either increase or decrease the stratification near the coast owing to coastal up- and downwelling responses (with offshore and onshore Ekman transport in the upper layer) to alongshore wind depending on its direction. The water column in the coastal area can be either re-stratified (downwelling favorable wind with equatorward flow, Fig. R1 left) or homogenized (upwelling favorable wind, Fig. R1 right) depending on the alongshore wind.

Figure R1. Schematics of isopycnals or isotherms (dashed line) and alongshore currents at the upper layer in response to downwelling (left) and upwelling (right) favorable wind stress.

Line 176. 'a high surface CF zone in the northern area'. Is it a general feature in summer only? Why CF is high in the northern area? In general, the summer rainfall is much higher than that of the other seasons in the Korean peninsula and freshwater discharged from the rivers increases as previously reported (Bae et al., 2008; Kong et al., 2013). We believe that the nutrient loading associated with the river discharges in the northern coastal areas trigger blooms in summer, as often seen from relatively high GOCI CF in areas nearby river mouths (e.g., JJ, WS, SP, and HH).

Line 237. '4.2. Other mechanisms'. It would be better to add more discussion. More discussions were added in Section 4.2 of the revised manuscript as recommended by this and other reviewers.

Lines 275, 281, 286. Check 'Park et al., 2018'. Park and Nam, 2018 ? This has been revised as "Park and Nam (2018)".

Line 297, '80%, 8 of 10'. It may not be conclusive. We counted the number of events where the (equatorward) alongshore advection plays a primary role in changing the CF at ESROB. More specifically, equatorward currents and salinity decreases were accompanied during the 8 events (E01–E06, E08–E09). The high CFs during the remaining two events (E07 and E10) were discussed in association with cross-advections (both onshore and offshore advections). Thus, we concluded that all the summertime CF events at ESROB could be explained by the horizontal, not vertical advection, and local blooms were not triggered by biogeochemical mechanisms (nutrient loading or light availability).

Please also note the supplement to this comment:
https://www.biogeosciences-discuss.net/bg-2018-183/bg-2018-183-AC2-supplement.pdf

---

## Author Response (AR2)

**Response to reviewer's second comments on "Summertime episodic chlorophyll-a blooms**
**near the east coast of Korea" by Y. -T. Son, J. -H. Park, and S. H. Nam**

The authors would like to thank the editor and reviewer for careful and constructive comments.
We have responded to the reviewer's comments (written in black) as below in blue.

___________________________________________

Reviewer #1

Review of: Summer-time episodic chlorophyll-a blooms near east coast of Korea
by Young-Tae Son, Jae-Hyoung Park, and SungHyun Nam
Manuscript ID: BG 2018 183 (2nd submission)
General comments
This paper has greatly improved and I thank the Authors for the effort.

I suggest some minor revisions below, after which I deem the paper to be ready for publication. Details
can be found below, and the Authors can contact me via the Editor, anytime for any questions. I do have
a scientific question for the Authors, NOT part of the review, but maybe for future work, if they want.
If you compare salinity in Fig. 6 with the rest, you find out that the summer 2012 at ESROB is way
more saltier. Are the authors planning to find out why? Also, in general, the salinity drops in summer
but not always in the same way. E.g. mid-June to mid-August in 2011 (Fig. 5), then there is NO DROP
in 2012 (Fig. 6) and from mid-July to end of record beyond September in 2013 (Fig. 7). Why this
variability in salinity drop period? And also why this salinity drop at all? From the figures it doesn't
look as if it's due to rainfall. Anyway this is just a curiosity.

Thank you for the questions and interest. We appreciate the reviewer's constructive comments. There
are surely interesting features observed at ESROB and not detailed as a focus in this paper, which
include the interannual variation in the summer-mean alongshore current and water properties such as
salinity near the coast. Yes, the salinity variations are hardly explained solely by the local precipitation
(rainfall) minus evaporation. We are currently analyzing the long-term (~16 years) time-series data
collected at ESROB, and more than happy to discuss the results with this reviewer, beyond this paper.
Our preliminary results suggest that the equatorward propagating coastal-trapped waves (CTWs),
forced primarily by remote winds off the Russian coast, along with influence of cross-shore movement
of the poleward-flowing offshore current (EKWC), play a decisive role in the interannual anomalies of
the alongshore current.

The English of the manuscript is now correct, with a few little additional corrections (see below).
Particular comments and suggested text corrections

Title, Abstract and 1. Introduction
OK.

Data and Methods

Line 62. "upper-most" -> "uppermost"
We have revised the word following the suggestion.

Line 86. "at a position 50 times higher than previous polar orbiting ocean color satellites" -> "and its altitude is 50 times higher (35,786 km) than that of polar orbiting ocean color satellites". GOCI is not polar orbiting because it's goo-stationary, so non need for "previous", I guess.
We have revised the phrase following the suggestion.

Line 90. "applying" -> "by applying"
We have revised the word following the suggestion.

Line 95. "variation for spatial Chl a distribution" -> "variation in Chl a spatial distribution"
We have revised the phrase following the suggestion.

Line 98. "by Korea" -> "by the Korea"
We have revised the word following the suggestion.

3. Results
3.1. Climatological CF variations

Lines 21-22 and Fig. 3. "Weak poleward... prevailed". From Fig. 3, how do you explain that, in Jan-May and Oct-Dec, alongshore currents are poleward (weakly positive) when alongshore wind stress is equatorward (negative)? Is the current not wind-driven in these months? (This is not mandatory, but interesting to have a comment).
It is reported from Park et al. (2016) that an equatorward reversal of coastal current in summer opposes poleward local wind stress and offshore current (EKWC). They suggest, by analyzing long time-series data collected at ESROB and other supplementary data, that the alongshore buoyancy gradient driven by the wind curl gradient and the prevalence of warmer and lower salinity water of northern origin balances the lateral friction.

3.2. CF events observed in summers of 2011, 2012, and 2013

Line 125-6. "where the surface CF was significantly" -> "by the surface CF being significantly".
We have revised the phrase following the suggestion.

Line 126-7. "were defined as a period of" -> "were defined as the period during which"
We have revised the phrase following the suggestion.

Line 146. "(showing downwelling induced by equatorward wind stress)" -> "(a consequence of downwelling induced by the equatorward wind stress, as also testified by the rise in temperature at all levels before e04, in Fig. 6 d)". I wrote "the rise in T", but if you have a better idea for proof, please write it in its place. That is, I think you should (easily) prove with your data that you have downwelling, because the equatorward the current is not a definite proof, though it goes in the good direction.

We have confirmed the temperature increase and revised the phrase following the suggestion.

3.3. Surface CF distributions

Line 175. "reached to JJ" -> "reached JJ"
We have revised the word following the suggestion.

Line 175. "farther south near the coast by July 27" -> "most probably hit ESROB by July 27 (Fig. 8d is
cloudy, but salinity drops in Fig. 7)"
We have revised the phrase following the suggestion.

Line 179. "Interestingly" -> "Coherently with this picture,"
We have revised the word following the suggestion.

Line 183. "main axis" -> "its main axis"
We have revised the word following the suggestion.

Line 189. "The offshore advection of coastal plume water of northern origin..." For the E10 event, Fig.
7 shows near-zero currents at ESROB. Also, salinity doesn't show a wedge drop. Also, Fig. 9c shows a
cyclonic eddy offshore of SO-BGN-DH. So how can one be sure that E10 is not due to recirculation of
southern CF-rich water? I think it could be either or both northern OR southern CF-rich water. Chl
imagery would solve the problem, but is not available, buit the lack of a wedge drop in SSS is maybe a
telltale that it is an E07-like event (even though summer salinity is in general much lower than that at
E07).
We agree that E10 'may' be affected by the CF-rich water of either or both northern and southern origins.
Although we do not rule out the possibility the reviewer suggested with no available GOCI CF image,
we retain the description (retaining 'may') raising the other possibility based on 1) equatorward
advection of the coastal plume water or northern origin having low salinity, high temperature, and high
CF (Fig. 7), and 2) dominant offshore current near DH and ESROB (Fig. 9c).

Discussion

4.1. Horizontal advection

Line 197. "8 of 10" -> "8 out of 10"
We have revised the word following the suggestion.

Line 215. "transportation" -> "transport"
We have revised the word following the suggestion.

Line 224. "half the" -> "half of the"
We have revised the word following the suggestion.

Lne 226 -> "northern origin" -> "northern origin, similarly to E09 and other northern water advection events," (just to give a little more coherence among the "classes" of events you talk about, i.e. northern
vs. southern water advection events).
We have revised the phrase following the suggestion.

4.2. Other mechanisms
Line 244. "nutrient supplied" -> "nutrients supplied"
We have revised the word following the suggestion.

Line 245. "by surface" -> "by the surface"
We have revised the word following the suggestion.

Line 251. "averaged" -> "average"
We have revised the word following the suggestion.

4.3. Inter-annual variations
Line 267. "in and around" -> "to and around"
We have revised the word following the suggestion.

Line 268. "year-to-year" -> "from year to year" (maybe better)
We have revised the word following the suggestion.

Line 273. "half the" -> "half of the"
We have revised the word following the suggestion.

Line 277. "coastal trapped" -> "coastally trapped"
We have retained the original word as widely used by the community.

Line 285. "impact of EKWC" -> "role of the EKWC"
We have revised the word following the suggestion.

Line 286. "yields less" -> "reveals a reduced"
We have revised the word following the suggestion.

Line 286. "in 2011" -> "during the 2011"
We have revised the word following the suggestion.

Line 287. "than 2012" -> "than in the 2012"
We have revised the word following the suggestion.

5. Concluding remarks

Line 293. "northern coast" -> "northern Korean coast"
We have revised the word following the suggestion.

Line 301. "play" -> "plays"
We have revised the word following the suggestion.

Tables, Figures and captions
Line 404. Figure 1 caption. "rainfall station" -> "rainfall stations"
We have revised the word following the suggestion.

Line 413. Figure 2 caption. "(dash lines)" -> "(dashed lines)"
We have revised the word following the suggestion.

Figure 3. The labels in the rectangles of each panel (E.g. WSa, WSc in the top plot) are virtually
unreadable, especially subscripts, also at a 120% zoom, please enlarge characters. Lines are OK.
The characters in the labels have been enlarged.

Line 423. Figure 4 caption. "percentage saturation" -> "percent saturation"
We have revised the word following the suggestion.

Line 479. Figure 10 caption. "colorbar in the right" -> "colorbar on the right".

We have revised the word following the suggestion.

**Summertime episodic chlorophyll-a blooms near the east coast of Korea**

Young-Tae Son, Jae-Hyoung Park, and SungHyun Nam[*]

School of the Earth and Environmental Sciences/Research Institute of Oceanography, Seoul National University, Seoul 08826, 
[revised manuscript text omitted]
 (a consequence of downwelling induced by the equatorward wind stress, as also testified by the rise in temperature at all levels before E04, Fig. 6d) developed before and during the most of E04 were weakened, and

SSS increased to reduce the salinity stratification and decrease surface CF during E04 (arrow labeled by K in Fig.

6b, c, e, and f). After the typhoon passed, the surface CF increased again along with re-enhancing equatorward currents, re-stratifying salinity, and decreasing SSS during E05 (Fig. 6c, e, and f). Two typhoons (BOLAVEN and

TENBIIN) successively passed the area and poleward (equatorward) wind stress imposed by BOLAVEN

(TENBIIN) induced an upwelling (downwelling) response with poleward (equatorward) and offshore (onshore)

transports at the upper layer, decreasing (increasing) water temperature, and increasing (decreasing) salinity in the whole column during E06. The poleward wind stress imposed by the BOLAVEN induced well-mixed conditions with high SSS, low SST, and strong poleward surface currents (arrow labeled by B in Fig. 6b, c, d, and f). However, the reversed wind stress imposed by the successive TENBIN resulted in decreasing SSS, increasing SST, weakening the poleward surface current (strengthening equatorward surface current), and rapidly increasing surface CF (peak exceeding 4.5 μg/$l$) (arrow labeled by T in Fig. 6b, c, d, e, and f).

Contrasting to the CF bloom events in the summers of 2011 and 2012, two among the four events (E07 and E10)

in the summer of 2013 did not follow heavy enough rainfall at the upstream stations nor equatorward currents (Fig. 7a, f). Typical heavy rainfall and enhanced equatorward surface currents preceded low SSS and high surface

CF during the other two events (E08 and E09) only (Fig. 7a, f). Unlikely with typical events, the SSS remained high and SST temporally decreased (negative anomaly) during E07 (Fig. 7c and d), whereas relatively high SST

and low SSS were observed during E10 (Fig. 7c, d). Contrasting with the other two years, winds were mild, and no typhoon passage was reported in the summer of 2013 (Fig. 7b).

**3.3. Surface CF distributions**

The equatorward advection of low salinity, chlorophyll-rich plume water into the ESROB area along the coast was confirmed from a series of daily composite GOCI Chl a only when clear images containing few clouds were available. One example presented here is from four images continuously available from July 24 to 27, 2013, before

E09 (Fig. 8a–d). A high surface CF zone in the northern area (e.g., off the SP, HH, and WS sites, Fig. 1) was separated from that in the southern area (e.g., between the coast and UI, Fig. 1) following the poleward current—

the East Korea Warm Current (EKWC)—whereas a more coastal branch extended equatorward with time near the coast (Fig. 8a–d) after the heavy rainfall of July 19–24 (Fig. 7a). The high CF plume water was elongated and reached  JJ by July 24, SO by July 25–26, and most probably hit ESROB by July 27

[revised manuscript text omitted]